# PolySHAP: Extending KernelSHAP with Interaction-Informed Polynomial Regression

**Fabian Fumagalli**
Bielefeld University
ffumagalli@techfak.de

**R. Teal Witter**
Claremont McKenna College
rtealwitter@cmc.edu

**Christopher Musco**
New York University
cmusco@nyu.edu

## Abstract

Shapley values have emerged as a central game-theoretic tool in explainable AI (XAI). However, computing Shapley values exactly requires $2^d$ game evaluations for a model with $d$ features. Lundberg and Lee's KernelSHAP algorithm has emerged as a leading method for avoiding this exponential cost. KernelSHAP approximates Shapley values by approximating the game as a linear function, which is fit using a small number of game evaluations for random feature subsets.

In this work, we extend KernelSHAP by approximating the game via higher degree polynomials, which capture non-linear interactions between features. Our resulting PolySHAP method yields empirically better Shapley value estimates for various benchmark datasets, and we prove that these estimates are consistent.

Moreover, we connect our approach to *paired sampling* (antithetic sampling), a ubiquitous modification to KernelSHAP that improves empirical accuracy. We prove that paired sampling outputs *exactly* the same Shapley value approximations as second-order PolySHAP, *without ever fitting a degree 2 polynomial.* To the best of our knowledge, this finding provides the first strong theoretical justification for the excellent practical performance of the paired sampling heuristic.

## 1 Introduction

Understanding the contribution of individual features to a model's prediction is a central goal in explainable artificial intelligence (XAI) (Covert & Lee, 2021). Among the most influential approaches are those grounded in cooperative game theory, where the *Shapley value* (Shapley, 1953) provides a principled way to distribute a model's output to its $d$ inputs.

The intuition behind the use of Shapley values is to attribute larger values to the players of a cooperative game with the most effect on the game's value. In XAI applications, players are typically features or training data points and the game value is typically a prediction or model loss.

Formally, we represent a cooperative game involving players $D = \{1, \dots, d\}$ via a value function $\nu : 2^D \to \mathbb{R}$ that maps subsets of players to values ($2^D$ denotes the powerset of $D$). Shapley values are then defined[1] via the best linear approximation to the game $\nu$. Concretely, for a subset $S \subseteq D$, $\nu(S)$ is approximated by a linear function in the binary features $\mathbb{1}[i \in S]$ for $i \in D$. The Shapley values are the coefficients of the linear approximation minimizing a specific weighted $\ell_2$ loss:

$$\phi^{\text{SV}}[\nu] := \underset{\phi \in \mathbb{R}^d : \langle \phi, \mathbf{1} \rangle = \nu(D)}{\arg \min} \sum_{S \subseteq D} \mu(S) \left( \nu(S) - \sum_{i=1}^{d} \phi_i \mathbb{1}[i \in S] \right)^2 ,$$

where the non-negative Shapley weights $\mu(S)$ are given in Equation (2). The constraint that the Shapley values sum to $\nu(D)$ enforces what is known as the "efficiency property", one of four axiomatic properties that motivate the original definition of Shapley values (see e.g., (Molnar, 2024)).

Since the sum above involves $2^d$ terms, exact minimization of the linear approximation to obtain $\phi^{\text{SV}}[\nu]$ is infeasible for most practical games. Over the past several years, substantial research has

---

[1]Without loss of generality, we assume $\nu(\emptyset) = 0$. Otherwise, we could consider the centered game $\nu(S) - \nu(\emptyset)$ which has the same Shapley values.

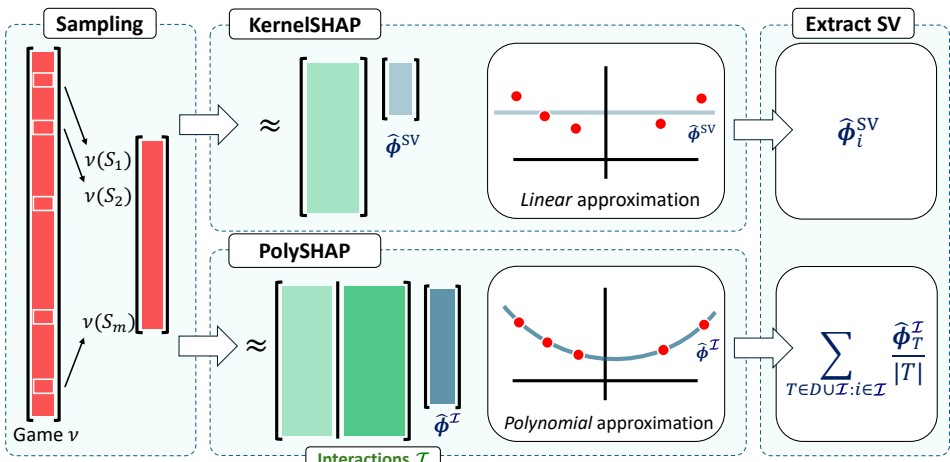

Figure 1: Both KernelSHAP and PolySHAP fit a function to approximate a sample of game evaluations. While KernelSHAP uses a linear approximation, PolySHAP uses a more expressive polynomial approximation. Finally, both algorithms return the Shapley values (SV) of their respective approximations (trivial for KernelSHAP, see Theorem 4.3 for PolySHAP).

focused on making the computation of Shapley values feasible in practice (Covert et al., 2020; Covert & Lee, 2021; Mitchell et al., 2022; Musco & Witter, 2025; Witter et al., 2025), with *KernelSHAP* (Lundberg & Lee, 2017) emerging as one of the most widely used model-agnostic methods.

From the least squares definition of Shapley values, KernelSHAP can be viewed as a two step process: First, approximate the game $\nu$ with a linear function fit from a sample of game evaluations $\nu(S)$ on randomly selected subsets $S$. Second, return the Shapley values of the approximation, which, for linear functions, are simply the coefficients of each input.

A natural idea is to adapt this framework to fit $\nu$ with a richer function class that still admits fast Shapley value computation. One such class is tree-based models like XGBoost, which Witter et al. (2025) recently leveraged to approximate the game. When the tree-based approximation is accurate, their RegressionMSR estimator produces more accurate Shapley value estimates than KernelSHAP. Butler et al. (2025) also use tree-based models to learn an approximation to the game, but extract Fourier coefficients which can be used to estimate more general attribution values. In the small sample regime with budgets less than $5d$, their ProxySPEX estimator outperforms KernelSHAP but achieves comparable performance to KernelSHAP for higher budgets.

In this work, we introduce an alternative approach called PolySHAP, where we approximate $\nu$ via a higher degree polynomial in the features $\mathbb{1}[i \in S]$ for $i \in D$, illustrated in Figure 1. For a degree $k$ polynomial, let $d' = O(d^k)$ be the number of terms. We show that, after fitting an approximation with $m$ samples, we can recover the Shapley values of the approximation in just $O(dd')$ time. Across various experiments, we find that higher degree PolySHAP approximations result in more accurate Shapley value estimates (see e.g., Figure 2). Moreover, we prove that the PolySHAP estimates are consistent, concretely that we obtain the Shapley values exactly as $m$ goes to $2^d$. This is in contrast to RegressionMSR, which needs an additional "regression adjustment" step to obtain a consistent estimator for tree-based approximations (Witter et al., 2025).

As a second main contribution of our work, we provide theoretical grounding for a seemingly unrelated sampling strategy called *paired sampling*, which is known to significantly improve the accuracy of KernelSHAP estimates (Covert & Lee, 2021; Mitchell et al., 2022; Olsen et al., 2024). In paired sampling, subsets are sampled in paired complements $S$ and $D \setminus S$. While used in all state-of-the-art Shapley value estimators, the reason for paired sampling's superior performance is not well understood. Surprisingly, we prove that KernelSHAP with paired sampling outputs *exactly* the same Shapley value approximations as second-order PolySHAP *without ever fitting a degree 2 polynomial*. This theoretical finding generalizes a very recent result of Mayer & Wüthrich (2025), who showed that KernelSHAP with paired sampling exactly recovers Shapley values when the game

has interactions of at most degree 2. Because the second-order PolySHAP will exactly fit a degree 2 game, their result follows immediately from a special case of ours. However, our finding is more general because it explains why paired sampling is effective for *all* games, not just those with at most degree 2 interactions.

**Contributions.** The main contributions of our work can be summarized as follows:

- We propose *PolySHAP*, an extension of KernelSHAP that models higher-order interaction terms to approximate $\nu$, and prove it returns the Shapley values as the number of samples $m$ goes to $2^d$ (Theorem 4.3). Moreover, we empirically show that PolySHAP results in more accurate Shapley value estimates than KernelSHAP and Permutation sampling.

- We establish a theoretical equivalence between paired KernelSHAP and second-order PolySHAP (Theorem 5.1), thereby explaining the practical benefits of paired sampling.

## 2 RELATED WORK

**KernelSHAP Sampling Strategies.** Prior work on improving KernelSHAP has focused on refining the subset sampling procedure, aiming to reduce variance and improve computational efficiency (Kelodjou et al., 2024; Olsen & Jullum, 2024; Musco & Witter, 2025). Among these enhancements, paired sampling produces the largest improvement in accuracy (Covert & Lee, 2021), yet, until the present work, it was not understood why beyond limited special cases. Another notable enhancement is in the sampling distribution. While it is intuitive to sample subsets proportional to their Shapley weights (Equation 2), it turns out that sampling proportional to the *leverage scores* can be more effective (Musco & Witter, 2025). Paired sampling has also been observed to improve LeverageSHAP (KernelSHAP with leverage score sampling).

**Other Shapley Value Estimators.** Beyond the regression-based approach of KernelSHAP, prior Shapley value estimators are generally based on direct Monte Carlo approximation (Kwon & Zou, 2022a; Castro et al., 2009; Kwon & Zou, 2022b; Kolpaczki et al., 2024; Li & Yu, 2024). These methods estimate the $i$-th Shapley value based on the following equivalent definition:

$$\phi_i^{\text{SV}}[\nu] = \frac{1}{d} \sum_{S \subseteq D \setminus \{i\}} \frac{\nu(S \cup \{i\}) - \nu(S)}{\binom{d-1}{|S|}}. \tag{1}$$

Permutation sampling, where subsets are sampled from a permutation, is a particularly effective approach (Castro et al., 2009; Mitchell et al., 2022). However, in direct Monte Carlo methods, each game evaluation is used to estimate at most two Shapley values. MSR methods reuse game evaluations in the estimate of every Shapley value, but at the cost of higher variance (Li & Yu, 2024; Witter et al., 2025). A recent benchmark finds that RegressionMSR with tree-based approximations, LeverageSHAP, KernelSHAP, and Permutation sampling are the most accurate (Witter et al., 2025).

**Higher-Order Explanations.** Another line of work seeks to improve approximations by explicitly modeling higher-order interactions. $k_{\text{ADD}}$-SHAP (Pelegrina et al., 2023) solves a least-squares problem over all interactions up to order $k$, and converges to the Shapley value for $k = 2$ and $k = 3$ (Pelegrina et al., 2025). With our results, we simplify $k_{\text{ADD}}$-SHAP and prove general convergence, where the practical differences are discussed in Appendix A.4. Relatedly, Mohammadi et al. (2025) propose a regularized least squares method based on the Möbius transform (Rota, 1964), which converges only when all higher-order interactions are included. By contrast, PolySHAP converges for any chosen set of interaction terms. Beyond approximation of the Shapley value, Kang et al. (2024) leverage the Fourier representation of games to detect and quantify higher-order interactions.

## 3 PRELIMINARIES ON EXPLAINABLE AI AND COOPERATIVE GAMES

**Notation.** We use boldface letters to denote vectors, e.g., $\boldsymbol{x}$, with entries $x_i$, and the corresponding random variable $\tilde{\boldsymbol{x}}$. The all-one vector is denoted by $\mathbf{1}$, and $\langle \cdot, \cdot \rangle$ is the standard inner product.

Given the prediction of a machine learning model $f : \mathbb{R}^d \to \mathbb{R}$, post-hoc feature-based explanations aim to quantify the contribution of features $D$ to the model output. Such explanations are defined by (i) the choice of an explanation game $\nu : 2^D \to \mathbb{R}$ and (ii) a game-theoretic attribution measure,

such as the Shapley value (Covert et al., 2021). For a given instance $\boldsymbol{x} \in \mathbb{R}^d$, the *local explanation game* $\nu_{\boldsymbol{x}}$ describes the model's prediction when restricted to subsets of features, with the remaining features replaced through perturbation. The perturbation is carried out using different imputation strategies, as summarized in Table 1.

Table 1: Local explanation games $\nu_{\boldsymbol{x}}$ for instance $\boldsymbol{x}$.

| Method | Game | $\nu_{\boldsymbol{x}}(S)$ | $\nu_{\boldsymbol{x}}(\emptyset)$ |
|---|---|---|---|
| **Baseline** | $\nu_{\boldsymbol{x}}^{(b)}$ | $f(\boldsymbol{x}_S, \boldsymbol{b}_{D \setminus S})$ | $f(\boldsymbol{b})$ |
| **Marginal** | $\nu_{\boldsymbol{x}}^{(m)}$ | $\mathbb{E}[f(\boldsymbol{x}_S, \tilde{\boldsymbol{x}}_{D \setminus S})]$ | $\mathbb{E}[f(\tilde{\boldsymbol{x}})]$ |
| **Conditional** | $\nu_{\boldsymbol{x}}^{(c)}$ | $\mathbb{E}[f(\tilde{\boldsymbol{x}}) \mid \tilde{\boldsymbol{x}}_S = \boldsymbol{x}_S]$ | $\mathbb{E}[f(\tilde{\boldsymbol{x}})]$ |

Similarly, *global explanation games* are constructed from $\nu_{\boldsymbol{x}}$ by evaluating measures such as variance or risk (Fumagalli et al., 2025). Beyond analyzing features, other variants have been proposed, for instance to characterize properties of individual data points (Ghorbani & Zou, 2019).

Like most Shapley value estimators, PolySHAP is agnostic to how the game $\nu$ is defined.

**KernelSHAP.** Given a budget of $m$ game evaluations, KernelSHAP solves the approximate least squares problem:

$$\hat{\boldsymbol{\phi}}^{\text{SV}}[\nu] := \underset{\boldsymbol{\phi} \in \mathbb{R}^d : \langle \boldsymbol{\phi}, \mathbf{1} \rangle = \nu(D)}{\arg \min} \sum_{\ell=1}^{m} \frac{\mu(S_\ell)}{p(S_\ell)} \left( \nu(S_\ell) - \sum_{i=1}^{d} \phi_i \mathbb{1}[i \in S_\ell] \right)^2 \quad \text{with } S_1, \dots, S_m \sim p$$

where the Shapley weight, for subset $S \subseteq D$ is given by

$$\mu(S) := \frac{1}{\binom{d-2}{|S|-1}} \quad \text{if } 0 < |S| < d \quad \text{and } 0 \text{ otherwise.} \tag{2}$$

While effective, KernelSHAP is inherently limited to a linear (additive) approximation of $\nu$ based on the sampled coalitions.

## 4 INTERACTION-INFORMED APPROXIMATION OF SHAPLEY VALUES

We now present *PolySHAP*, our core contribution that extends KernelSHAP by approximating the game via polynomial regression. Rather than fitting a linear function, PolySHAP captures higher-order feature interactions, leading to more accurate Shapley value estimates. We first introduce the PolySHAP representation, then discuss sampling strategies and interaction frontier construction.

### 4.1 POLYSHAP INTERACTION REPRESENTATION

We introduce PolySHAP, a method for producing Shapley value estimates from a polynomial approximation of $\nu$. Let the **interaction frontier** $\mathcal{I}$ be a subset of interaction terms

$$\mathcal{I} \subseteq \{T \subseteq D : |T| \geq 2\}.$$

We then extend the linear approximation of $\nu$ by defining an *interaction-based polynomial representation* restricted to interactions in $\mathcal{I}$.

**Definition 4.1.** *The **PolySHAP representation** $\boldsymbol{\phi}^{\mathcal{I}} \in \mathbb{R}^{d'}$ with $d' = d + |\mathcal{I}|$ is given by*

$$\boldsymbol{\phi}^{\mathcal{I}}[\nu] := \underset{\boldsymbol{\phi} \in \mathbb{R}^{d'} : \langle \boldsymbol{\phi}, \mathbf{1} \rangle = \nu(D)}{\arg \min} \sum_{S \subseteq D} \mu(S) \left( \nu(S) - \sum_{T \in D \cup \mathcal{I}} \phi_T \prod_{j \in T} \mathbb{1}[j \in S] \right)^2.$$

*Here, and in the following, we abuse notation with $\phi_i := \phi_{\{i\}}$ and $\mathbb{1}[j \in i] := \mathbb{1}[j = i]$ for $i, j \in D$.*

The PolySHAP representation generalizes the least squares formulation of the Shapley value to arbitrary interaction frontiers $\mathcal{I}$. For each interaction set $T \in \mathcal{I}$, the approximation contributes a coefficient $\phi_T$ only if all features in $T$ are present in $S$.

**Remark 4.2.** *The PolySHAP representation directly extends the Faithful Shapley interaction index ([Tsai et al., 2023](#)) to arbitrary interaction frontiers.*

In the theorem below, we show how to recover Shapley values from the PolySHAP representation.

**Theorem 4.3.** *The Shapley values of $\nu$ are recovered from the PolySHAP representation as*

$$\phi_i^{SV}[\nu] = \phi_i^{\mathcal{I}} + \sum_{S \in \mathcal{I}: i \in S} \frac{\phi_S^{\mathcal{I}}}{|S|} \quad for \ i \in D. \tag{3}$$

In other words, consistent estimation of the PolySHAP representation directly implies consistent estimation of the Shapley value.

## 4.2 POLYSHAP ALGORITHM

A natural approximation strategy is to first estimate the PolySHAP representation and then map the result back to Shapley values using Theorem 4.3. Concretely, we approximate the PolySHAP representation by solving

$$\hat{\phi}^{\mathcal{I}}[\nu] := \underset{\phi \in \mathbb{R}^{d+|\mathcal{I}|} : \langle \phi, \mathbf{1} \rangle = \nu(D)}{\arg\min} \sum_{\ell=1}^{m} \frac{\mu(S_\ell)}{p(S_\ell)} \left( \nu(S_\ell) - \sum_{T \in D \cup \mathcal{I}} \phi_T \prod_{j \in T} \mathbb{1}[j \in S_\ell] \right)^2 \tag{4}$$

with $m$ samples $S_1, \ldots, S_m$ drawn from some distribution $p$, where $d' < m \leq 2^d$. (When $\nu$ is clear from context, we write $\hat{\phi}^{\mathcal{I}}$ for $\hat{\phi}^{\mathcal{I}}[\nu]$.)

We then convert $\hat{\phi}^{\mathcal{I}}$ into Shapley value estimates via Theorem 4.3. The rationale behind this approach is that the more expressive PolySHAP representation more accurately represents $\nu$, which in turn yields more accurate Shapley value estimates. We refer to this interaction-aware extension of KernelSHAP as *PolySHAP*.

In order to produce the PolySHAP solution in practice, we use the matrix representation of the regression problem. Define the sampled design matrix $\tilde{\mathbf{X}} \in \mathbb{R}^{m \times d'}$ and the sampled target vector $\tilde{\mathbf{y}} \in \mathbb{R}^m$. The rows are indexed by $\ell \in [m]$, and the columns of $\tilde{\mathbf{X}}$ are indexed by interactions $T \in D \cup \mathcal{I}$. The entries of the sampled design matrix and sampled target vector are given by

$$[\tilde{\mathbf{X}}]_{\ell,T} = \sqrt{\frac{\mu(S_\ell)}{p(S_\ell)}} \cdot \mathbb{1}[T \subseteq S_\ell] \quad \text{and} \quad [\tilde{\mathbf{y}}]_\ell = \sqrt{\frac{\mu(S_\ell)}{p(S_\ell)}} \cdot \nu(S_\ell). \tag{5}$$

In this notation, we may write

$$\hat{\phi}^{\mathcal{I}} = \underset{\phi \in \mathbb{R}^{d'} : \langle \mathbf{1}, \phi \rangle = \nu(D)}{\arg\min} \|\tilde{\mathbf{X}}\phi - \tilde{\mathbf{y}}\|_2^2. \tag{6}$$

We would like to apply standard regression tools when solving the problem, so we convert from the constrained problem to an unconstrained reformulation. Let $\mathbf{P}_{d'}$ be the matrix that projects *off* the all ones vector in $d'$ dimensions i.e., $\mathbf{P}_{d'} = \mathbf{I} - \frac{1}{d'}\mathbf{1}_{d'}\mathbf{1}_{d'}^\top$. We have

$$\underset{\phi \in \mathbb{R}^{d'} : \langle \mathbf{1}, \phi \rangle = \nu(D)}{\arg\min} \|\tilde{\mathbf{X}}\phi - \tilde{\mathbf{y}}\|_2^2 = \underset{\phi \in \mathbb{R}^{d'} : \langle \phi, \mathbf{1} \rangle = 0}{\arg\min} \left\| \tilde{\mathbf{X}}\phi + \tilde{\mathbf{X}}\mathbf{1}\frac{\nu(D)}{d'} - \tilde{\mathbf{y}} \right\|_2^2 + \mathbf{1}\frac{\nu(D)}{d'}$$

$$= \mathbf{P}_{d'} \underset{\phi \in \mathbb{R}^{d'}}{\arg\min} \left\| \tilde{\mathbf{X}}\mathbf{P}_{d'}\phi + \tilde{\mathbf{X}}\mathbf{1}\frac{\nu(D)}{d'} - \tilde{\mathbf{y}} \right\|_2^2 + \mathbf{1}\frac{\nu(D)}{d'}. \tag{7}$$

PolySHAP is described in pseudocode in Algorithm 1.

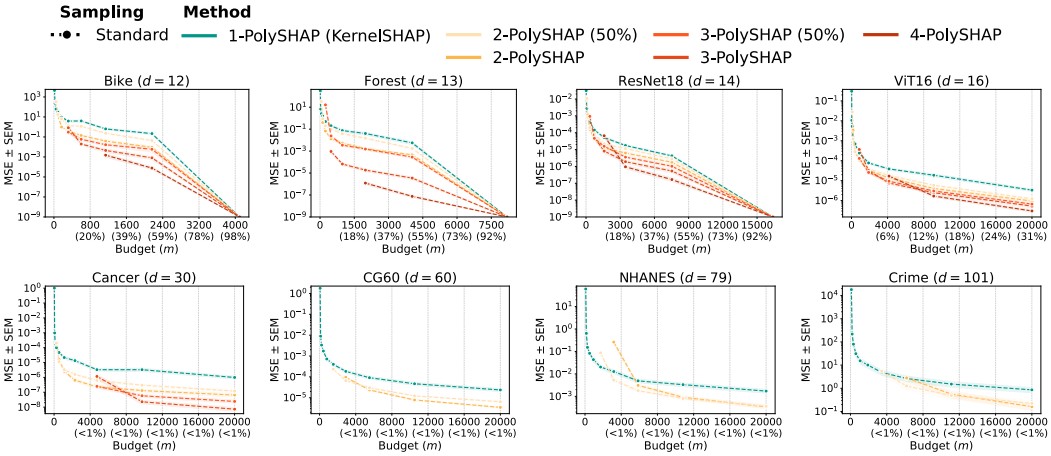

Figure 2: Approximation quality measured by MSE ($\pm$ SEM) for various sampling budgets $m$ on different games. Adding any number of interactions in PolySHAP improves approximation quality.

---

**Algorithm 1** PolySHAP

---

**Require:** game $\nu_{\mathbf{x}}$, interaction frontier $\mathcal{I}$, sampling distribution $p$, sampling budget $m > d'$.
1: Define $\nu(S) := \nu_{\mathbf{x}}(S) - \nu_{\mathbf{x}}(\emptyset)$       ▷ Center for notational simplicity
2: $\{S_\ell\}_{\ell=1}^m \leftarrow \text{SAMPLE}(m, p)$
3: Construct $\tilde{\mathbf{X}}$ and $\tilde{\mathbf{y}}$       ▷ Equation (5)
4: $\hat{\boldsymbol{\phi}}^{\mathcal{I}} \leftarrow \mathbf{P}_{d'} \text{SOLVELEASTSQUARES}(\tilde{\mathbf{X}}\mathbf{P}_{d'}, \tilde{\mathbf{y}} - \tilde{\mathbf{X}}\mathbf{1}\frac{\nu(D)}{d'}) + \mathbf{1}\frac{\nu(D)}{d'}$
5: $\hat{\boldsymbol{\phi}}^{\text{SV}} \leftarrow \text{POLYSHAPTOSV}(\hat{\boldsymbol{\phi}}^{\mathcal{I}})$       ▷ Equation (3)
6: **return** $\nu_{\boldsymbol{x}}(\emptyset), \hat{\boldsymbol{\phi}}^{\text{SV}}$

---

**Computational Complexity.** The computational complexity of PolySHAP can be divided into two components: evaluating the game for the sampled coalitions, and solving the regression problem followed by extraction of the Shapley values. Evaluating the game requires at least one model call for local explanation games, and highly depends on the application setting. Solving the regression problem scales with $\mathcal{O}(m \cdot (d')^2 + (d')^3)$, whereas transforming the PolySHAP representation to Shapley values is of order $\mathcal{O}(d \cdot d')$. Importantly, this complexity scales *linearly* with the budget $m$, and *quadratically* with the number of regression variables $d'$. In practice, the dominant factor in computational cost is usually the game evaluations, i.e., the model predictions. However, for smaller model architectures, the runtime can be influenced by the number of regression variables.

## 4.3 Sampling Strategies for PolySHAP

PolySHAP uses a distribution $p$ to sample $m$ game evaluations for approximating the least squares objective. Previous work (Lundberg & Lee, 2017; Covert & Lee, 2021) chose $p$ proportional to $\mu(S)$, which cancels the multiplicative correction term in Equation (4).

However, sampling proportionally to *leverage scores* offers improved estimation quality, and is supported by theoretical guarantees (Musco & Witter, 2025). Let $\mathbf{X} \in \mathbb{R}^{2^d \times d'}$ be the full deterministic matrix (each subset is sampled exactly once with probability 1). The leverage score for the row corresponding to subset $S$ is given by

$$\ell_S = [\mathbf{X}\mathbf{P}_{d'}]_S^\top \left(\mathbf{P}_{d'}\mathbf{X}^\top\mathbf{X}\mathbf{P}_{d'}\right)^\dagger [\mathbf{X}\mathbf{P}_{d'}]_S \qquad (8)$$

where $(\cdot)^\dagger$ denotes the pseudoinverse, and $[\mathbf{X}\mathbf{P}_{d'}]_S$ is the $S$-th row of $\mathbf{X}\mathbf{P}_{d'}$.

**Theorem 4.4** (Leverage Score Sampling Guarantee (Musco & Witter, 2025))**.** *Let $\epsilon, \delta > 0$. When $m = O(d' \log \frac{d'}{\delta} + d' \frac{1}{\epsilon\delta})$ subsets are sampled proportionally to their leverage scores (with or without*

*replacement and with or without paired sampling), $\hat{\phi}^{\mathcal{I}}$ satisfies, with probability $1 - \delta$,*

$$\sum_{S \subseteq D} \mu(S) \left( \nu(S) - \sum_{T \in D \cup \mathcal{I}} \hat{\phi}_T^{\mathcal{I}} \prod_{j \in T} \mathbb{1}[j \in S] \right)^2$$

$$\leq (1 + \epsilon) \sum_{S \subseteq D} \mu(S) \left( \nu(S) - \sum_{T \in D \cup \mathcal{I}} \phi_T^{\mathcal{I}} \prod_{j \in T} \mathbb{1}[j \in S] \right)^2.$$

Musco & Witter (2025) show that $\ell_S = 1/\binom{d}{|S|}$ for KernelSHAP, i.e., leverage score sampling is equivalent to sampling subsets uniformly by their size. For the $k$-additive interaction frontier (introduced below), we can directly compute the leverage scores using symmetry and Equation (8), although a closed-form solution remains unknown. In practice, we observed little variation between leverage scores of order 1 and those of higher orders, which is why we recommend using order-1 leverage scores.

## 4.4 Construction of Interaction Frontiers

The interaction frontier $\mathcal{I}$ determines the number of additional variables (columns) in the linear regression problem. Its size must be balanced against the budget $m$ (rows). Since lower-order interaction terms occur more frequently and are thus less sensitive to noise, it is natural to expand these terms first. To this end, we define the $k$-**additive interaction frontier** for $k = 2, \ldots, d$ as

$$\mathcal{I}_{\leq k} := \{S \subseteq D : 2 \leq |S| \leq k\} \quad \text{with } |\mathcal{I}_{\leq k}| = \sum_{i=2}^{k} \binom{d}{i}.$$

The $k$-additive interaction frontier includes all interactions up to order $k$ by sequentially extending the $(k-1)$-additive interaction frontier with $\binom{d}{k}$ sets. It is widely used in Shapley-based interaction indices (Sundararajan et al., 2020; Tsai et al., 2023; Bordt & von Luxburg, 2023). In the following, we refer to PolySHAP using $\mathcal{I}_{\leq k}$ as **k-PolySHAP**.

**Corollary 4.5.** *The $k$-PolySHAP representation is equal to order-$k$ Faith-SHAP (Tsai et al., 2023).*

A notable special case of $k$-PolySHAP is the interaction frontier without interactions: 1-PolySHAP, i.e., without interactions ($\mathcal{I} = \emptyset$), is equivalent to KernelSHAP.

We further show convergence for $k_{\text{ADD}}$-SHAP, extending Theorem 4.2 in Pelegrina et al. (2025).

**Proposition 4.6.** *$k_{ADD}$-SHAP converges to the Shapley value for $k = 1, \ldots, d$.*

$k_{\text{ADD}}$-SHAP is linked to $k$-PolySHAP, but we recommend PolySHAP in practice, see Appendix A.4.

**Partial Interaction Frontiers.** In high dimensions, the $k$-additive interaction frontier grows combinatorially with $\binom{d}{k}$. With a limited evaluation budget $m$, including all interaction terms of a given order may yield an underdetermined least-squares system. To address this, we introduce the **partial interaction frontier** $\mathcal{I}_\ell$ with exactly $\ell$ elements:

$$\mathcal{I}_\ell := \mathcal{I}_{\leq k_\ell} \cup \mathcal{R}, \quad \text{with } |\mathcal{I}_\ell| = \ell,$$

where $k_\ell$ is the largest order such that $|\mathcal{I}_{\leq k_\ell}| \leq \ell$, and $\mathcal{R} \subseteq \mathcal{I}_{\leq k_{\ell+1}} \setminus \mathcal{I}_{\leq k_\ell}$ denotes a set of $\ell - |\mathcal{I}_{\leq k_\ell}|$ interaction terms of order $k_\ell + 1$. In words, $\mathcal{I}_\ell$ sequentially covers the $k$-additive interaction frontier up to $k_\ell$, and supplements it with a selected subset of the subsequent higher-order interactions. In our experiments, we demonstrate that partially including higher-order interactions improves approximation quality, whereas using the full $k$-additive interaction frontier provides the largest gains.

## 5 Paired KernelSHAP is Paired 2-PolySHAP

A common heuristic when estimating Shapley values is to sample subsets in pairs $S$ and $D \setminus S$. A kind of antithetic sampling (Glasserman, 2004), paired sampling substantially improves the approximation of estimators (Covert & Lee, 2021; Mitchell et al., 2022; Olsen & Jullum, 2024). Adding

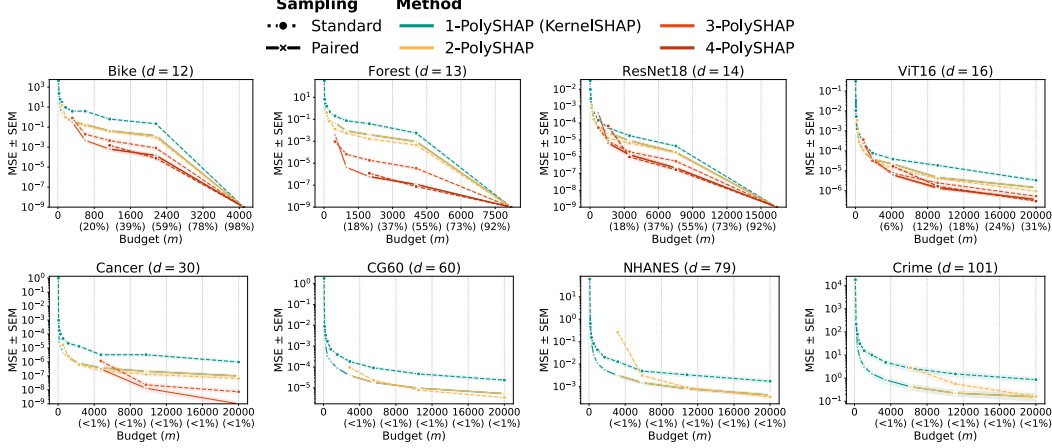

Figure 3: Approximation quality measured by MSE ($\pm$ SEM) for standard (dotted) and paired (solid) sampling. With paired sampling, KernelSHAP achieves the same performance as 2-PolySHAP.

higher-order interactions to PolySHAP improves Shapley value estimates, provided we have enough samples (see Figure 2): 3-PolySHAP outperforms 2-PolySHAP, which outperforms KernelSHAP (1-PolySHAP). Surprisingly, paired sampling partially collapses this hierarchy (see Figure 3).

**Theorem 5.1** (Paired KernelSHAP is Paired 2-PolySHAP). *Suppose that subsets are sampled in pairs i.e., if $S$ is sampled then so is its complement $D \setminus S$, and, the matrix $\tilde{\mathbf{X}}$ has full column rank for interaction frontier $D$ and $\mathcal{I}_{\leq 2}$. Then*

$$\hat{\phi}^{SV} = \text{POLYSHAPTOSV}(\hat{\phi}^{\mathcal{I}_{\leq 2}})$$

*In words, Shapley values approximated by 2-PolySHAP are precisely the KernelSHAP estimates.*

We prove Theorem 5.1 by explicitly building the approximate solutions of KernelSHAP and 2-PolySHAP. Of particular help is a new technical projection lemma that we also use in the proof of Theorem 4.3. See Appendix A for the details.

**Generalizing Prior Work.** Mayer & Wüthrich (2025) recently showed that paired KernelSHAP exactly recovers the Shapley values of games with interactions of at most size 2. This follows immediately from Theorem 5.1, because 2-PolySHAP will precisely fit a game with order-2 interactions and paired KernelSHAP will return the same solution. However, Theorem 5.1 is far more general because it explains why paired sampling performs so well for *all* games, not just a restricted class.

**Higher Dimensional Extensions.** A natural question is whether similar results hold for higher-order interactions. Suppose $k$ is an odd number, we find empirically that paired $(k+1)$-PolySHAP returns the same approximate Shapley values as paired $k$-PolySHAP. We conjecture that this pattern holds for all odd $k$ such that $1 \leq k < d$. However, it is not obvious how to adapt our proof of Theorem 5.1, since we would need the explicit mapping of $k + 1$-PolySHAP representations to $k$-PolySHAP representations (this is clear when $k = 1$, but not so for higher dimensions).

## 6 EXPERIMENTS

We empirically validate PolySHAP and approximate Shapley values on 15 local explanation games across 30 randomly selected instances, see Table 2. We evaluate all methods with $m$ samples ranging from $d + 1$ to $\min(2^d, 20000)$, and compare PolySHAP against *Permutation Sampling* (Castro et al., 2009), *SVARM* (Kolpaczki et al., 2024), *MSR* (Fumagalli et al., 2023; Wang & Jia, 2023), *Unbiased KernelSHAP* (Covert & Lee, 2021), *RegressionMSR* with XGBoost (Witter et al., 2025), and *KernelSHAP* (1-PolySHAP) with leverage score sampling (Lundberg & Lee, 2017; Musco & Witter, 2025).

For tabular datasets, we trained random forests, while for image classification we used a `ResNet18` (He et al., 2016) with 14 superpixels and vision transformers with $3\times3$ (ViT9) and $4\times4$ (ViT16) super-patches on ImageNet (Deng et al., 2009), and CIFAR-10 (Krizhevsky et al., 2009). For language modeling, we used a fine-tuned `DistilBert` (Sanh et al., 2019) to predict sentiment on the IMDB dataset (Maas et al., 2011) with review excerpts of length 14. For tabular datasets, the games were defined via path-dependent feature perturbation, allowing ground-truth Shapley values to be obtained from TreeSHAP (Lundberg et al., 2020). For all other datasets, we used baseline imputation and exhaustive Shapley value computation. As evaluation metrics, we report *mean-squared error (MSE)*, top-5 precision (*Precision@5*), and *Spearman correlation* with *standard error of the mean (SEM)*. Code is available at `https://github.com/FFmgll/PolySHAP`, and additional details and results, including a runtime analysis, are provided in Appendix B.

**PolySHAP Variants.** For comparability across methods, we sample subsets using order-1 leverage scores, i.e., uniformly over subset sizes. We further adopt sampling without replacement and distinguish between standard and paired subset sampling. We apply the *border trick* (Fumagalli et al., 2023), replacing random sampling with exhaustive enumeration of sizes when the expected samples exceed the number of subsets. We use *k-PolySHAP* with $k \in \{1, 2, 3, 4\}$, and additionally the partial interaction frontiers that cover $50\%$ of all $k$-order interactions, denoted by *k-PolySHAP (50%)*. For high-dimensional settings, we introduce PolySHAP (log) that adds $d \log(\binom{d}{3})$ order-3 interactions.

**Higher-order Interactions Improve Approximation.** Figure 2 reports the MSE with SEM for selected explanation games and standard sampling. Across different games, we observe that incorporating higher-order interactions in PolySHAP consistently improves approximation quality. However, higher-order PolySHAP requires a larger sampling budget, and hence performance is only plotted for $m \geq d'$. Nevertheless, 2-PolySHAP, and even partial interaction inclusion (e.g., 2-PolySHAP at 50%), still yield notable improvements in approximation accuracy.

Table 2: Explanation games.

| ID | $d$ | Domain |
|---|---|---|
| Housing | 8 | tabular |
| ViT9 | 9 | image |
| Bike | 12 | tabular |
| Forest | 13 | tabular |
| Adult | 14 | tabular |
| ResNet18 | 14 | image |
| DistilBERT | 14 | language |
| Estate | 15 | tabular |
| ViT16 | 16 | image |
| CIFAR10 | 16 | image |
| Cancer | 30 | tabular |
| CG60 | 60 | synthetic |
| IL60 | 60 | synthetic |
| NHANES | 79 | tabular |
| Crime | 101 | tabular |

**Paired KernelSHAP is 2-PolySHAP.** As shown in Theorem 5.1, under paired sampling, KernelSHAP and 2-PolySHAP are equivalent indicated by the overlapping lines. We confirm this empirically in Figure 3. However, there is an important distinction: 2-PolySHAP requires more budget, whereas KernelSHAP can be computed already with $d + 1$ samples. Lastly, we observe a similar pattern for 3-PolySHAP: Under paired sampling 3-PolySHAP substantially improves its approximation quality and is equivalent to 4-PolySHAP.

**Practical Benefits of PolySHAP.** In practice, we adopt paired sampling and benchmark PolySHAP against all baselines in Figure 4 and Figure 7 in Appendix B.2. Because of our paired sampling result, the practical benefits of PolySHAP become apparent only when order-3 interactions are included. In low-dimensional settings, the 3-PolySHAP yields the best performance on Housing, Adult, Estate, Forest, and Cancer datasets (see e.g., Figure 4 and Figure 7). In budget-restricted cases, partially incorporating order-3 interactions already provides substantial gains, cf. 3-PolySHAP (50%) and 3-PolySHAP (log). In high-dimensional settings ($d \geq 60$), however, only a small number of order-3 interactions can be added, resulting in more modest improvements. Among all baselines, only RegressionMSR achieves comparable performance, although its performance depends strongly on XGBoost, as indicated by its poor results on CG60. Moreover, RegressionMSR has an inherent advantage since all tabular games rely on tree-based models.

# 7 CONCLUSION & FUTURE WORK

By reformulating the computation of the Shapley value as a polynomial regression problem with selected interaction terms, PolySHAP extends beyond the linear regression framework of KernelSHAP. We demonstrate that PolySHAP provides consistent estimates of the Shapley value (The-

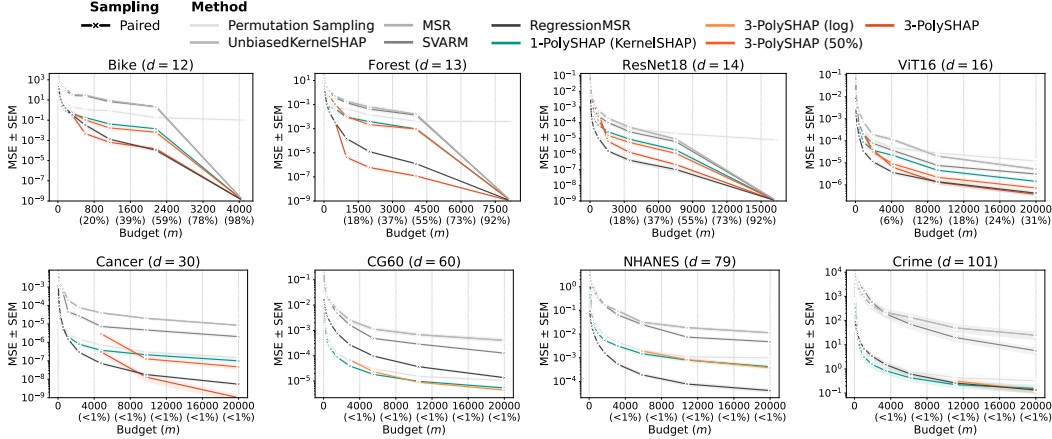

Figure 4: Approximation quality of PolySHAP variants and baseline methods measured by MSE ($\pm$ SEM) using paired sampling. With paired sampling, PolySHAP consistently improves upon KernelSHAP when order-3 interactions are included. In higher dimensions ($d \geq 60$), only a few of these can be modeled, yielding smaller improvements.

orem 4.3), and produces more accurate Shapley value estimates (see Figure 2 and Figure 3). Moreover, we show that paired subset sampling in KernelSHAP (Covert & Lee, 2021) implicitly captures all second-order interactions at no extra cost (Theorem 5.1), explaining why paired sampling improves estimator accuracy on games with arbitrary interaction structures.

Future work could explore more structured variants of interaction frontier, for example by detecting important interactions (Tsang et al., 2020) or leveraging inherent interaction structures in graph-structured inputs (Muschalik et al., 2025). In addition, we empirically find that paired $k$-PolySHAP produces the same estimates as $(k + 1)$-PolySHAP for odd $k > 1$, but leave the proof for future work.

## ETHICS STATEMENT

This work introduces a framework for efficient approximation of Shapley values, which are primarily used for explainable AI. We do not see any ethical concerns associated with this work.

## REPRODUCIBILITY STATEMENT

Our implementation is built upon the `shapiq` (Muschalik et al., 2024) library. We extend this framework with the `PolySHAP` and `RegressionMSR` approximator classes, which include optimized sampling strategies within the `CoalitionSampler` class. All technical details can be found in Section 6 and Appendix B. The code for reproducing the experiments is available at https://github.com/FFmgll/PolySHAP. We will soon add our methods to `shapiq`.

## ACKNOWLEDGMENTS

Fabian Fumagalli gratefully acknowledges funding by the Deutsche Forschungsgemeinschaft (DFG, German Research Foundation): TRR 318/3 2026 – 438445824. Christopher Musco was supported by NSF Award #2045590.

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

CONTENTS OF THE APPENDIX

# A   PROOFS

## A.1   PROJECTION LEMMA

We introduce the following technical lemma that will be useful in the proofs of Theorem 4.3 and Theorem 5.1.

**Lemma A.1** (Projection Lemma). *Let $n \geq d_+ > d$. Consider a matrix $\mathbf{X} \in \mathbb{R}^{n \times d}$ with full column rank, a vector $\mathbf{y} \in \mathbb{R}^n$, and a real number $c \in \mathbb{R}$. Let $\mathbf{X}_+ \in \mathbb{R}^{n \times d_+}$ be a matrix where the first $d$ columns are equal to $\mathbf{X}$. Define*

$$\boldsymbol{\beta}_+^* = \underset{\boldsymbol{\beta} \in \mathbb{R}^{d_+} : \langle \boldsymbol{\beta}, \mathbf{1}_{d_+} \rangle = c}{\arg\min} \|\mathbf{X}_+ \boldsymbol{\beta} - \mathbf{y}\|_2^2.$$

*Then*

$$\underset{\boldsymbol{\beta} \in \mathbb{R}^d : \langle \boldsymbol{\beta}, \mathbf{1} \rangle = c}{\arg\min} \|\mathbf{X}\boldsymbol{\beta} - \mathbf{y}\|_2^2 = \underset{\boldsymbol{\beta} \in \mathbb{R}^d : \langle \boldsymbol{\beta}, \mathbf{1}_d \rangle = c}{\arg\min} \|\mathbf{X}\boldsymbol{\beta} - \mathbf{X}_+ \boldsymbol{\beta}_+^*\|_2^2. \tag{9}$$

*Proof of Lemma A.1.* We will first reformulate the constrained least squares problem as an unconstrained problem. Let $\mathbf{P}_d$ be the matrix that projects *off* the all ones vector in $d$ dimensions i.e., $\mathbf{P}_d = \mathbf{I} - \frac{1}{d} \mathbf{1}_d \mathbf{1}_d^\top$. Similarly, let $\mathbf{P}_{d_+} = \mathbf{I} - \frac{1}{d_+} \mathbf{1}_{d_+} \mathbf{1}_{d_+}^\top$. In general, we will drop the subscript $d$ when the dimension is clear from context. We have

$$\underset{\boldsymbol{\beta} \in \mathbb{R}^d : \langle \boldsymbol{\beta}, \mathbf{1} \rangle = c}{\arg\min} \|\mathbf{X}\boldsymbol{\beta} - \mathbf{y}\|_2^2 = \underset{\boldsymbol{\beta} \in \mathbb{R}^d : \langle \boldsymbol{\beta}, \mathbf{1} \rangle = 0}{\arg\min} \left\| \mathbf{X}\boldsymbol{\beta} + \mathbf{X}\mathbf{1}\frac{c}{d} - \mathbf{y} \right\|_2^2 + \mathbf{1}\frac{c}{d}$$

$$= \mathbf{P}_d \underset{\boldsymbol{\beta} \in \mathbb{R}^d}{\arg\min} \left\| \mathbf{X}\mathbf{P}_d\boldsymbol{\beta} + \mathbf{X}\mathbf{1}\frac{c}{d} - \mathbf{y} \right\|_2^2 + \mathbf{1}\frac{c}{d}$$

$$= \mathbf{P}_d (\mathbf{X}\mathbf{P}_d)^\dagger \left( \mathbf{y} - \mathbf{X}\mathbf{1}\frac{c}{d} \right) + \mathbf{1}\frac{c}{d}, \tag{10}$$

where the $(\cdot)^\dagger$ denotes the pseudoinverse, and the last equality follows by the standard solution to an unconstrained least squares problem. Similarly,

$$\boldsymbol{\beta}_+^* = \underset{\boldsymbol{\beta} \in \mathbb{R}^{d_+} : \langle \boldsymbol{\beta}, \mathbf{1}_{d_+} \rangle = c}{\arg\min} \|\mathbf{X}_+ \boldsymbol{\beta} - \mathbf{y}\|_2^2 = \mathbf{P}_{d_+} (\mathbf{X}_+ \mathbf{P}_{d_+})^\dagger \left( \mathbf{y} - \mathbf{X}_+ \mathbf{1}\frac{c}{d_+} \right) + \mathbf{1}\frac{c}{d_+}. \tag{11}$$

Let $\mathbf{proj}_{\mathbf{X}\mathbf{P}_d} = (\mathbf{X}\mathbf{P}_d)(\mathbf{X}\mathbf{P}_d)^\dagger$ be the projection *onto* $\mathbf{X}\mathbf{P}_d$. We have

$$\mathbf{X} \underset{\boldsymbol{\beta} \in \mathbb{R}^d : \langle \boldsymbol{\beta}, \mathbf{1} \rangle = c}{\arg\min} \|\mathbf{X}\boldsymbol{\beta} - \mathbf{X}_+ \boldsymbol{\beta}_+^*\|_2^2 = \mathbf{X}\mathbf{P}_d(\mathbf{X}\mathbf{P}_d)^\dagger \left( \mathbf{X}_+ \boldsymbol{\beta}_+^* - \mathbf{X}\mathbf{1}\frac{c}{d} \right) + \mathbf{X}\mathbf{1}\frac{c}{d}$$

$$= \mathbf{proj}_{\mathbf{X}\mathbf{P}_d} \left( \mathbf{X}_+ \left[ \mathbf{P}_{d_+} (\mathbf{X}_+ \mathbf{P}_{d_+})^\dagger \left( \mathbf{y} - \mathbf{X}_+ \mathbf{1}\frac{c}{d_+} \right) + \mathbf{1}\frac{c}{d_+} \right] - \mathbf{X}\mathbf{1}\frac{c}{d} \right) + \mathbf{X}\mathbf{1}\frac{c}{d}$$

$$= \mathbf{proj}_{\mathbf{X}\mathbf{P}_d}\mathbf{proj}_{\mathbf{X}_+ \mathbf{P}_{d_+}} \mathbf{y} - \textcolor{red}{\mathbf{proj}_{\mathbf{X}\mathbf{P}_d}\mathbf{proj}_{\mathbf{X}_+ \mathbf{P}_{d_+}} \mathbf{X}_+ \mathbf{1}\frac{c}{d_+}} + \textcolor{red}{\mathbf{proj}_{\mathbf{X}\mathbf{P}_d}\mathbf{X}_+ \mathbf{1}\frac{c}{d_+}}$$

$$- \mathbf{proj}_{\mathbf{X}\mathbf{P}_d}\mathbf{X}\mathbf{1}\frac{c}{d} + \mathbf{X}\mathbf{1}\frac{c}{d}. \tag{12}$$

Since the column space of $\mathbf{X}\mathbf{P}_d$ is contained in the column space of $\mathbf{X}_+ \mathbf{P}_{d_+}$, observe that $\mathbf{proj}_{\mathbf{X}\mathbf{P}_d}\mathbf{proj}_{\mathbf{X}_+ \mathbf{P}_{d_+}} = \mathbf{proj}_{\mathbf{X}\mathbf{P}_d}$. Then

$$(12) = \mathbf{proj}_{\mathbf{X}\mathbf{P}_d}\mathbf{y} - \mathbf{proj}_{\mathbf{X}\mathbf{P}_d}\mathbf{X}\mathbf{1}\frac{c}{d} + \mathbf{X}\mathbf{1}\frac{c}{d}$$

$$= \mathbf{X}\mathbf{P}_d(\mathbf{X}\mathbf{P}_d)^\dagger \left( \mathbf{y} - \mathbf{X}\mathbf{1}\frac{c}{d} \right) + \mathbf{X}\mathbf{1}\frac{c}{d}$$

$$= \mathbf{X} \underset{\boldsymbol{\beta} \in \mathbb{R}^d : \langle \boldsymbol{\beta}, \mathbf{1} \rangle = c}{\arg\min} \|\mathbf{X}\boldsymbol{\beta} - \mathbf{y}\|_2^2. \tag{13}$$

Since $\mathbf{X}$ has full column rank, we have $\mathbf{X}^\dagger \mathbf{X} = \mathbf{I}$, so multiplying on the left by $\mathbf{X}^\dagger$ yields the statement.

$\square$

## A.2 PolySHAP is Consistent

In this section, we will prove Theorem 4.3.

**Theorem 4.3.** *The Shapley values of $\nu$ are recovered from the PolySHAP representation as*

$$\phi_i^{SV}[\nu] = \phi_i^{\mathcal{I}} + \sum_{S \in \mathcal{I}: i \in S} \frac{\phi_S^{\mathcal{I}}}{|S|} \quad \text{for } i \in D. \tag{3}$$

*Proof of Theorem 4.3.* Recall $d' = d + |\mathcal{I}|$. Define the target vector $\mathbf{y} \in \mathbb{R}^{2^d}$ so that $[\tilde{\mathbf{y}}]_S = \sqrt{\mu(S)} \cdot \nu(S_\ell)$. Define the design matrix $\mathbf{X} \in \mathbb{R}^{2^d \times d}$ so that

$$[\mathbf{X}]_{S,i} = \sqrt{\mu(S)} \cdot \mathbb{1}[i \in S], \tag{14}$$

and the extended design matrix $\mathbf{X}_+ \in \mathbb{R}^{2^d \times d'}$ so that

$$[\mathbf{X}_+]_{S,T} = \sqrt{\mu(S)} \cdot \mathbb{1}[T \subseteq S], \tag{15}$$

for $T \in D \cup \mathcal{I}$.

In this notation, we may write

$$\boldsymbol{\phi}^{SV}[\nu] = \underset{\boldsymbol{\phi} \in \mathbb{R}^d : \langle \mathbf{1}, \boldsymbol{\phi} \rangle = \nu(D)}{\arg\min} \|\mathbf{X}\boldsymbol{\phi} - \mathbf{y}\|_2^2, \tag{16}$$

and

$$\boldsymbol{\phi}^{\mathcal{I}}[\nu] = \underset{\boldsymbol{\phi} \in \mathbb{R}^{d'} : \langle \mathbf{1}, \boldsymbol{\phi} \rangle = \nu(D)}{\arg\min} \|\mathbf{X}_+\boldsymbol{\phi} - \mathbf{y}\|_2^2. \tag{17}$$

Consider the game $\hat{\nu} : 2^D \to \mathbb{R}$ where

$$\hat{\nu}(S) = \sum_{T \in D \cup \mathcal{I}: T \subseteq S} \phi_T^{\mathcal{I}}[\nu]. \tag{18}$$

For this game, the target vector is given by $\hat{\mathbf{y}} = \mathbf{X}_+ \boldsymbol{\phi}^{\mathcal{I}}[\nu]$. Then its Shapley values are given by

$$
\begin{aligned}
\boldsymbol{\phi}^{SV}[\hat{\nu}] &= \underset{\boldsymbol{\phi} \in \mathbb{R}^d : \langle \mathbf{1}, \boldsymbol{\phi} \rangle = \nu(D)}{\arg\min} \|\mathbf{X}\boldsymbol{\phi} - \hat{\mathbf{y}}\|_2^2 \\
&= \underset{\boldsymbol{\phi} \in \mathbb{R}^d : \langle \mathbf{1}, \boldsymbol{\phi} \rangle = \nu(D)}{\arg\min} \|\mathbf{X}\boldsymbol{\phi} - \mathbf{X}_+\boldsymbol{\phi}^{\mathcal{I}}\|_2^2 \\
&= \underset{\boldsymbol{\phi} \in \mathbb{R}^d : \langle \mathbf{1}, \boldsymbol{\phi} \rangle = \nu(D)}{\arg\min} \|\mathbf{X}\boldsymbol{\phi} - \mathbf{y}\|_2^2 \\
&= \boldsymbol{\phi}^{SV}[\nu],
\end{aligned} \tag{19}
$$

where the penultimate equality follows by Lemma A.1. All that remains is to compute the Shapley values $\hat{\nu}$. Since we have an explicit representation of $\hat{\nu}$ in terms of its Möbius transform in Equation (18), we know its Shapley values are

$$\phi_i^{SV}[\hat{\nu}] = \sum_{T \in D \cup \mathcal{I}: i \in T} \frac{\phi_T^{\mathcal{I}}[\nu]}{|T|}. \tag{20}$$

by e.g., Table 3 in Grabisch et al. (2000). The statement follows.

$\square$

### A.3 PAIRED KERNELSHAP IS PAIRED 2-POLYSHAP

We introduce some helpful notation, and then use it to restate Theorem 5.1 more formally below.

Define $d_k = \sum_{\ell=1}^k \binom{d}{\ell}$. Let $\tilde{\mathbf{X}}_k \in \mathbb{R}^{m \times \binom{d}{k}}$ be the matrix where the $\ell, T$ entry is given by

$$[\tilde{\mathbf{X}}_k]_{\ell,T} = \frac{\sqrt{\mu(S_\ell)}}{\sqrt{p(S_\ell)}} \mathbb{1}[T \subseteq S_\ell] \tag{21}$$

where $S_\ell \subseteq D$ is the $\ell$th sampled subset, and $T \subseteq D$ such that $|T| = k$. Then the matrix $\tilde{\mathbf{X}}_{\leq k} \in \mathbb{R}^{m \times d_k}$ is given by

$$\tilde{\mathbf{X}}_{\leq k} = \begin{bmatrix} \tilde{\mathbf{X}}_1 & \dots & \tilde{\mathbf{X}}_k \end{bmatrix}. \tag{22}$$

Let $\mathbf{M}_{2\to 1} \in \mathbb{R}^{d \times d_2}$ be the matrix that projects a 2-PolySHAP to a 1-PolySHAP. The entry corresponding to $i \in D$, and $S \subseteq D$ such that $|S| \leq 2$ is given by

$$[\mathbf{M}_{2\to 1}]_{i,S} = \frac{\mathbb{1}[i \in S]}{|S|}. \tag{23}$$

**Theorem A.2** (Paired KernelSHAP is Paired 2-PolySHAP). *Suppose $\tilde{\mathbf{X}}_{\leq 2}$ has full column rank. Further, suppose that both 1-PolySHAP and 2-PolySHAP are computed with the* same *paired samples i.e., if $S$ is sampled then so is its complement $D \setminus S$. Then*

$$\underset{\phi \in \mathbb{R}^d : \langle \mathbf{1}_d, \phi \rangle = \nu(D)}{\arg\min} \|\tilde{\mathbf{X}}_1 \phi - \tilde{\mathbf{y}}\|_2^2 = \mathbf{M}_{2\to 1} \underset{\phi \in \mathbb{R}^{d_2} : \langle \mathbf{1}_{d_2}, \phi \rangle = \nu(D)}{\arg\min} \|\tilde{\mathbf{X}}_{\leq 2} \phi - \tilde{\mathbf{y}}\|_2^2. \tag{24}$$

*In words, the Shapley values of the approximate 1-PolySHAP are exactly the same as those of the approximate 2-PolySHAP.*

*Proof of Theorem A.2.* Define

$$\tilde{\mathbf{z}} + \mathbf{1}_{d_2} \frac{\nu(D)}{d_2} = \underset{\phi \in \mathbb{R}^{d_2} : \langle \phi, \mathbf{1}_{d_2} \rangle = \nu(D)}{\arg\min} \|\tilde{\mathbf{X}}_{\leq 2} \phi - \tilde{\mathbf{y}}\|_2^2, \tag{25}$$

where $\tilde{\mathbf{z}}$ is orthogonal to the all ones vector. By Lemma A.1 and the structure $\mathbf{A}_{\leq 2} = \begin{bmatrix} \mathbf{A}_1 & \mathbf{A}_2 \end{bmatrix}$, we have

$$\underset{\phi \in \mathbb{R}^d : \langle \mathbf{1}_d, \phi \rangle = \nu(D)}{\arg\min} \|\tilde{\mathbf{X}}_1 \phi - \tilde{\mathbf{y}}\|_2^2 = \underset{\phi \in \mathbb{R}^d : \langle \mathbf{1}_d, \phi \rangle = \nu(D)}{\arg\min} \left\| \tilde{\mathbf{X}}_1 \phi - \tilde{\mathbf{X}}_{\leq 2} \left( \tilde{\mathbf{z}} + \mathbf{1}_{d_2} \frac{\nu(D)}{d_2} \right) \right\|_2^2. \tag{26}$$

Using Equation 10, we can write Equation 26 explicitly as

$$(26) = \mathbf{P}_d (\mathbf{P}_d \tilde{\mathbf{X}}_1^\top \tilde{\mathbf{X}}_1 \mathbf{P}_d)^\dagger \mathbf{P}_d \tilde{\mathbf{X}}_1^\top \left[ \tilde{\mathbf{X}}_{\leq 2} \left( \tilde{\mathbf{z}} + \mathbf{1}_{d_2} \frac{\nu(D)}{d_2} \right) - \tilde{\mathbf{X}}_1 \mathbf{1}_d \frac{\nu(D)}{d} \right] + \mathbf{1}_d \frac{\nu(D)}{d} \tag{27}$$

where $\mathbf{P}_d = \mathbf{1} - \frac{1}{d}\mathbf{1}\mathbf{1}^\top$ is the matrix that projects off the all ones direction in $d$ dimensions.

Our goal is to show that

$$(27) = \mathbf{M}_{2\to 1} \left( \tilde{\mathbf{z}} + \mathbf{1}_{d_2} \frac{\nu(D)}{d_2} \right). \tag{28}$$

We'll begin with the all ones component. Observe that

$$\frac{\nu(D)}{d} [\mathbf{M}_{2\to 1}\mathbf{1}]_i = \frac{\nu(D)}{d_2} \left( \sum_{j=1}^d \mathbb{1}[i = j] + \sum_{T \subseteq D : |T|=2} \frac{\mathbb{1}[i \in T]}{2} \right) = \nu(D) \frac{1 + \frac{d-1}{2}}{d + \binom{d}{2}} = \frac{\nu(D)}{d},$$

so $\mathbf{M}_{2\to 1} \mathbf{1}_{d_2} \frac{\nu(D)}{d_2} = \mathbf{1}_d \frac{\nu(D)}{d}$.

Now it remains to show the equality for the component orthogonal to the all ones direction. Since $\tilde{\mathbf{X}}_{\leq 2} = \begin{bmatrix} \tilde{\mathbf{X}}_1 & \tilde{\mathbf{X}}_2 \end{bmatrix}$ has full column rank by assumption, $\tilde{\mathbf{X}}_1$ must have full column rank as well. It

follows that $(\mathbf{P}_d \tilde{\mathbf{X}}_1^\top \tilde{\mathbf{X}}_1 \mathbf{P}_d)(\mathbf{P}_d \tilde{\mathbf{X}}_1^\top \tilde{\mathbf{X}}_1 \mathbf{P}_d)^\dagger = \mathbf{P}_d$. Then, after multiplying Equations 27 and 28 by $(\mathbf{P}_d \tilde{\mathbf{X}}_1^\top \tilde{\mathbf{X}}_1 \mathbf{P}_d)$, it suffices to show that

$$
\begin{aligned}
\mathbf{P}_d \tilde{\mathbf{X}}_1^\top \tilde{\mathbf{X}}_1 \mathbf{P}_d \mathbf{M}_{2\to 1} \tilde{\mathbf{z}} &= \mathbf{P}_d \tilde{\mathbf{X}}_1^\top \left[ \tilde{\mathbf{X}}_{\leq 2} \left( \tilde{\mathbf{z}} + \mathbf{1}_{d_2} \frac{\nu(D)}{d_2} \right) - \tilde{\mathbf{X}}_1 \mathbf{1}_d \frac{\nu(D)}{d} \right] \\
&= \mathbf{P}_d \tilde{\mathbf{X}}_1^\top \tilde{\mathbf{X}}_{\leq 2} \tilde{\mathbf{z}} + \mathbf{P}_d \left[ \tilde{\mathbf{X}}_1^\top \tilde{\mathbf{X}}_{\leq 2} \mathbf{1}_{d_2} \frac{\nu(D)}{d_2} - \tilde{\mathbf{X}}_1^\top \tilde{\mathbf{X}}_{\leq 1} \mathbf{1}_d \frac{\nu(D)}{d} \right]. \quad (29)
\end{aligned}
$$

We will first show that the second term on the right hand side is 0. First, notice that

$$
[\tilde{\mathbf{X}}_{\leq k} \mathbf{1}_{d_k}]_S = \sum_{T \subseteq D : |T| \leq k} \frac{\sqrt{\mu(S)}}{\sqrt{p(S)}} \mathbb{1}[T \subseteq S] = \frac{\sqrt{\mu(S)}}{\sqrt{p(S)}} |S|_k \quad (30)
$$

where $|S|_k = \sum_{\ell=1}^k \binom{|S|}{\ell}$. Then

$$
\frac{1}{d_k} [\tilde{\mathbf{X}}_1^\top \tilde{\mathbf{X}}_{\leq k} \mathbf{1}_{d_k}]_i = \sum_{S : i \in S} \frac{\mu(S)}{p(S)} \frac{|S|_k}{d_k}. \quad (31)
$$

We have $\frac{|S|_2}{d_2} = \frac{|S| + \binom{|S|}{2}}{d + \binom{d}{2}} = \frac{|S|}{d} \frac{1 + (|S|-1)/2}{1 + (d-1)/2} = \frac{|S|}{d} \cdot \frac{|S|+1}{d+1}$. Together,

$$
\begin{aligned}
\left[ \tilde{\mathbf{X}}_1^\top \tilde{\mathbf{X}}_{\leq 2} \mathbf{1}_{d_2} \frac{1}{d_2} - \tilde{\mathbf{X}}_1^\top \tilde{\mathbf{X}}_1 \mathbf{1}_d \frac{1}{d} \right]_i &= \sum_{S : i \in S} \frac{\mu(S)}{p(S)} \frac{|S|}{d} \left( \frac{|S|+1}{d+1} - \frac{d+1}{d+1} \right) \\
&= \frac{1}{d(d+1)} \sum_{S : i \in S} \frac{\mu(S)}{p(S)} |S|(|S|-d) \\
&= \frac{1}{2d(d+1)} \sum_S \frac{\mu(S)}{p(S)} |S|(|S|-d) \quad (32)
\end{aligned}
$$

where the last equality follows because the subsets are sampled in paired complements. In particular, for a given pair $S$ and $D \setminus S$, the item $i$ is in exactly one of them, and the coefficient $\frac{\mu(S)}{p(S)} |S|(|S|-d)$ is the same for both. We have shown that every entry is the same, i.e., a scaling of $\mathbf{1}$, so $\mathbf{P}_d$ projects off the entire vector.

Finally, it remains to show that

$$
\mathbf{P}_d \tilde{\mathbf{X}}_1^\top \tilde{\mathbf{X}}_1 \mathbf{P}_d \mathbf{M}_{2\to 1} \tilde{\mathbf{z}} = \mathbf{P}_d \tilde{\mathbf{X}}_1^\top \tilde{\mathbf{X}}_{\leq 2} \tilde{\mathbf{z}}. \quad (33)
$$

It is easy to verify that $\langle \mathbf{1}_d, \mathbf{M} \tilde{\mathbf{z}} \rangle = \langle \mathbf{1}_{d_2}, \tilde{\mathbf{z}} \rangle = 0$, so $\mathbf{P}_d \mathbf{M}_{2\to 1} \tilde{\mathbf{z}} = \mathbf{M}_{2\to 1} \tilde{\mathbf{z}}$. Therefore, it suffices to prove that $\mathbf{P}_d \tilde{\mathbf{X}}_1^\top \tilde{\mathbf{X}}_1 \mathbf{M}_{2\to 1} = \mathbf{P}_d \tilde{\mathbf{X}}_1^\top \tilde{\mathbf{X}}_{\leq 2}$.

Notice that $[\tilde{\mathbf{X}}_1^\top \tilde{\mathbf{X}}_1]_{i,j} = \sum_{S : i \in S, j \in S} \frac{\mu(S)}{p(S)}$ where $i, j \in D$. Then

$$
\begin{aligned}
[\tilde{\mathbf{X}}_1^\top \tilde{\mathbf{X}}_1 \mathbf{M}_{2\to 1}]_{i,R} &= \sum_{j=1}^d \frac{\mathbb{1}[j \in R]}{|R|} \sum_{S : i \in S, j \in S} \frac{\mu(S)}{p(S)} \\
&= \sum_{S : i \in S} \frac{\mu(S)}{p(S)} \sum_{\substack{j=1 \\ j \in S, j \in R}}^d \frac{1}{|R|} \\
&= \sum_{S : i \in S} \frac{\mu(S)}{p(S)} \frac{|R \cap S|}{|R|}. \quad (34)
\end{aligned}
$$

Meanwhile,

$$
[\tilde{\mathbf{X}}_1^\top \tilde{\mathbf{X}}_{\leq 2}]_{i,R} = \sum_{S : i \in S, R \subseteq S} \frac{\mu(S)}{p(S)}. \quad (35)
$$

Clearly, Equations (34) and (35) are equal when $|R| = 1$. Now consider the case when $|R| = 2$; we have

$$(34) = \sum_{S:i\in S,|R\cap S|=1} \frac{\mu(S)}{p(S)}\frac{1}{2} + \sum_{S:i\in S,|R\cap S|=2} \frac{\mu(S)}{p(S)} = \frac{1}{4}\sum_{S:|R\cap S|=1} \frac{\mu(S)}{p(S)} + \sum_{S:i\in S,R\subseteq S} \frac{\mu(S)}{p(S)}.$$

(36)

where the last equality follows by sampling in paired complements. In particular, exactly one of the paired samples $S$ and $D \setminus S$ will contain item $i$, and the coefficient $\mu(S)/p(S)$ is the same for both. Finally, because it is the same for all $i$, the projection $\mathbf{P}_d$ eliminates the first term. The statement follows.

$\square$

## A.4 $k$-ADD-SHAP CONVERGES TO THE SHAPLEY VALUE

In this section, we prove Proposition 4.6 and discuss the differences between PolySHAP and $k_{\text{ADD}}$-SHAP, and its practical implications. We generally recommend to prefer PolySHAP over $k_{\text{ADD}}$-SHAP.

**Proposition 4.6.** *$k_{ADD}$-SHAP converges to the Shapley value for $k = 1, \ldots, d$.*

*Proof.* The $k_{\text{ADD}}$ approximation algorithm (Pelegrina et al., 2023) is based on the interaction representation (Grabisch et al., 2000) of $\nu$ given by

$$\nu(S) = \sum_{T \subseteq D} \gamma_{|S \cap T|}^{|T|} I_{\text{Sh}}(T) \quad \text{with } \gamma_r^t := \sum_{\ell=0}^{r} \binom{r}{\ell} B_{t-\ell},$$

where $B_t$ are the Bernoulli numbers and $I_{\text{Sh}}$ is the Shapley interaction index (Grabisch & Roubens, 1999) with

$$I_{\text{Sh}}(S) := \sum_{T \subseteq D \setminus S} \frac{1}{(d - |S| + 1)\binom{d-|S|}{|T|}} \sum_{L \subseteq S} (-1)^{|S|-|L|} \nu(T \cup L).$$

The Shapley interaction index generalizes the Shapley value to arbitrary subsets, and it holds $\phi_i^{\text{SV}}[\nu] = I_{\text{Sh}}(i)$ for all $i \in D$. The $k_{\text{ADD}}$-SHAP approximation algorithm then restricts this representation to interactions up to order $k$.

**Definition A.3** ($k_{ADD}$-SHAP (Pelegrina et al., 2025)). *The $k_{ADD}$-SHAP algorithms solves the constrained weighted least-squares problem*

$$I^{k_{ADD}} := \underset{I \in \mathbb{R}^{\sum_{\ell=0}^{k} \binom{d}{\ell}}}{\arg\min} \sum_{S \subseteq D} \mu(S) \left( \nu(S) - \sum_{T \subseteq D: |T| \leq k} \gamma_{|S \cap T|}^{|T|} I_T \right)^2$$

$$s.t. \ \nu(D) - \nu(\emptyset) = \sum_{T \subseteq D: |T| \leq k} \left( \gamma_{|T|}^{|T|} - \gamma_0^{|T|} \right) I_T.$$

*In practice, the least-squares objective is approximated and solved similar to KernelSHAP (Lundberg & Lee, 2017), and the Shapley value estimates that are output are $I_i^{k_{ADD}}$ for $i \in D$ from the approximated least-squares system.*

Our first observation is that the output $I_i$ is the Shapley value of the approximated game, i.e.

$$\phi_i^{\text{SV}}\Big[ \sum_{T \subseteq D: |T| \leq k} \gamma_{\mathbb{1}[i \in T]}^{|T|} I_T \Big] = I_i.$$

We will show that the Shapley values of this approximation are the Shapley values of the PolySHAP representation $\phi^{\mathcal{I}_{\leq k}}$, which then are equal to the Shapley values of $\nu$ by Theorem 4.3.

In contrast to PolySHAP, $k_{\text{ADD}}$-SHAP fits a coefficient for the empty set $\phi_\emptyset$. However, we may rewrite

$$\left( \nu(S) - \sum_{T \subseteq D: |T| \leq k} \gamma_{|S \cap T|}^{|T|} I_T \right)^2 = \left( \nu(S) - \gamma_0^0 I_\emptyset - \sum_{T \subseteq D: 0 < |T| \leq k} \gamma_{|S \cap T|}^{|T|} I_T \right)^2,$$

and thus $I_\emptyset$ is an additive shift of $\nu$, which does not affect the Shapley values of the approximation, i.e.

$$\phi_i^{\text{SV}}\Big[ \sum_{T \subseteq D: |T| \leq k} \gamma_{|S \cap T|}^{|T|} I_T \Big] = \phi_i^{\text{SV}}\Big[ \sum_{T \subseteq D: 0 < |T| \leq k} \gamma_{|S \cap T|}^{|T|} I_T \Big].$$

Moreover, we can compute $\gamma_0^t = B_t$ and

$$\gamma_s^s = \sum_{\ell=0}^{s} \binom{s}{\ell} B_{s-\ell} = \sum_{\ell=0}^{s} \binom{s}{\ell} B_\ell = \sum_{\ell=0}^{s-1} \binom{s}{\ell} B_\ell + B_s = \mathbb{1}[s=1] + B_s,$$

by the recursion of Bernoulli numbers, and thus

$$\sum_{S \subseteq D : |S| \leq k} \left( \gamma_{|S|}^{|S|} - \gamma_0^{|S|} \right) I_S = \sum_{i \in D} I_i,$$

which is already mentioned by Pelegrina et al. (2025, Proof of Theorem 4.2). Now, without loss of generality, we can assume that $\nu(\emptyset) = 0$, since it does not affect the Shapley values of $\nu$, and thus the class of approximations is given by

$$\mathcal{F}^{k_{\text{ADD}}} := \left\{ S \mapsto \sum_{T \subseteq D : 0 < |T| \leq k} \gamma_{|S \cap T|}^{|T|} I_T : \phi \in \mathbb{R}^{d+|\mathcal{I}_{\leq k}|} \text{ and } \sum_{i \in D} I_i = \nu(D) \right\}.$$

**Lemma A.4.** *There is an equivalence between the function class $\mathcal{F}^{k_{\text{ADD}}}$ and the class of functions of PolySHAP representation with interaction frontier $\mathcal{I}_{\leq k}$, i.e.*

$$\mathcal{F}^{k_{\text{ADD}}} = \left\{ S \mapsto \sum_{T \in D \cup \mathcal{I}_{\leq k}} \phi_T \prod_{j \in T} \mathbb{1}[j \in S] : \phi \in \mathbb{R}^{d+|\mathcal{I}_{\leq k}|} \text{ and } \langle \phi, 1 \rangle = \nu(D) \right\}$$

*Proof.* For the game $\nu$ there exist the two equivalent representations (Grabisch et al., 2000, Table 3 and 4)

$$\nu(S) = \sum_{T \subseteq D} \gamma_{|S \cap T|}^{|T|} I_{\text{Sh}}(T) \quad \text{with } \gamma_r^t := \sum_{\ell=0}^{r} \binom{r}{\ell} B_{t-\ell},$$

where $I_{\text{Sh}}$ is the Shapley interaction index (Grabisch & Roubens, 1999), and the Möbius representation

$$\nu(S) = \sum_{T \subseteq D} m(T) \prod_{j \in T} \mathbb{1}[j \in S] \quad \text{with } m(S) := \sum_{L \subseteq S} (-1)^{|S|-|L|} \nu(L).$$

Moreover, there exist the two conversion formulas (Grabisch et al., 2000)[Table 3 and 4]

$$I_{\text{Sh}}(S) = \sum_{T \subseteq D : T \supseteq S} \frac{1}{|T| - |S| + 1} m(T) \text{ and } m(S) = \sum_{T \subseteq D : T \supseteq S} B_{|T|-|S|} I_{\text{Sh}}(T).$$

From the conversion formulas it is obvious that

$$I_{\text{Sh}}(S) = 0, \quad \forall S \subseteq D : |S| > k \quad \Leftrightarrow \quad m(S) = 0, \quad \forall S \subseteq D : |S| > k.$$

Hence, restricting the interaction representation to order $k$ yields the same function class as restricting the Möbius representation to order $k$. Moreover, the constraints are similarly converted, which proves the equivalence. $\square$

Utilizing Lemma A.4, we obtain that for

$$I^{k_{\text{ADD}}^+} := \underset{I \in \mathbb{R}^{\sum_{\ell=1}^{k} \binom{d}{\ell}}}{\arg \min} \sum_{S \subseteq D} \mu(S) \left( \nu(S) - \sum_{T \subseteq D : |T| \leq k} \gamma_{|S \cap T|}^{|T|} I_T \right)^2$$

$$\text{s.t. } \nu(D) = \sum_{i \in D} I_i$$

we have equivalence between the approximations

$$\sum_{T \subseteq D : |T| \leq k} \gamma_{|S \cap T|}^{|T|} I_T^{k_{\text{ADD}}^+} = \sum_{T \in D \cup \mathcal{I}_{\leq k}} \phi_T^{\mathcal{I}_{\leq k}} \prod_{j \in T} \mathbb{1}[j \in S],$$

where $\phi_T^{\mathcal{I}_{\leq k}}$ is the PolySHAP representation, due to the equivalent function classes parametrized by the vectors $I^{k_{\mathrm{ADD}}^+}$ and $\phi^{\mathcal{I}_{\leq k}}$. By Theorem 4.3, we know that the Shapley values of this approximation are equal to the Shapley values of $\nu$, and hence, we have

$$I_i^{k_{\mathrm{ADD}}} = \phi_i^{\mathrm{SV}}\Big[\sum_{T \subseteq D:|T| \leq k} \gamma_{|S \cap T|}^{|T|} I_T^{k_{\mathrm{ADD}}}\Big] = \phi_i^{\mathrm{SV}}\Big[\sum_{T \subseteq D:|T| \leq k} \gamma_{|S \cap T|}^{|T|} I_T^{k_{\mathrm{ADD}}^+}\Big] = \phi_i^{\mathrm{SV}}[\nu],$$

which concludes the proof and shows convergence of $k_{\mathrm{ADD}}$-SHAP to the Shapley value.

**Practical difference between $k_{\mathbf{ADD}}$-SHAP and PolySHAP.** In contrast to PolySHAP, $k_{\mathrm{ADD}}$-SHAP was proposed for $k$-additive interaction frontiers. Moreover, the design matrix of $k_{\mathrm{ADD}}$-SHAP is less intuitive, making the PolySHAP formulation a simpler and more transparent alternative. More importantly, a key practical difference arises from our use of the modified representation $I^{k_{\mathrm{ADD}}^+}$ as an intermediate step in the proof. While $I^{k_{\mathrm{ADD}}^+}$ and $I^{k_{\mathrm{ADD}}}$ yield the same Shapley values when all subsets are evaluated, they diverge under approximation. In particular, unlike PolySHAP, $k_{\mathrm{ADD}}$-SHAP is affected by the value of $\nu(\emptyset)$, and its least-squares fit includes an additional variable. For these reasons, we recommend PolySHAP in practice over $k_{\mathrm{ADD}}$-SHAP.

$\square$

Table 3: Datasets used for tabular explanation games

| Name (ID in bold) | Reference | License | Source |
|---|---|---|---|
| California **Housing** | (Kelley Pace & Barry, 1997) | Public Domain | `sklearn` |
| **Bike** Regression | (Fanaee-T & Gama, 2014) | CC-BY 4.0 | OpenML |
| **Forest** Fires | (Cortez & Morais, 2007) | CC-BY 4.0 | UCI Repo |
| **Adult** Census | (Kohavi, 1996) | CC-BY 4.0 | OpenML |
| Real **Estate** | (Yeh & Hsu, 2018) | CC-BY 4.0 | UCI Repo |
| Breast **Cancer** | (Street et al., 1993) | CC-BY 4.0 | `shap` |
| Correlated Groups (**CG60**) | synthetic | MIT | `shap` |
| Independent Linear (**IL60**) | synthetic | MIT | `shap` |
| **NHANES** I | (Dinh et al., 2019) | Public Domain | `shap` |
| Communities and **Crime** | (Redmond, 2011) | CC-BY 4.0 | `shap` |

## B  EXPERIMENTAL DETAILS AND ADDITIONAL RESULTS

In this section, we provide additional details regarding our experiments and the local explanation game setup (Appendix B.1) with additional results on the remaining games using MSE (Appendix B.2), Precision@5 (Appendix B.3), and Spearman correlation (Appendix B.4). Lastly, we report results of the runtime analysis (Appendix B.5).

### B.1  EXPERIMENTAL DETAILS

All experiments were conducted on a consumer-grade laptop with an 11th Gen Intel Core i7-11850H CPU and 30GB of RAM, where we used `cuda`[2] on a NVIDIA RTX A2000 GPU for inference of the CIFAR10 game.

**Non-tabular Datasets.** We used the 30 pre-computed games provided by the `shapiq` benchmark for the ResNET18 (He et al., 2016), and the vision transformers pre-trained on ImageNet (Deng et al., 2009). We used the pre-computed language game using a DistilBERT (Sanh et al., 2019) model and sentiment analysis on the IMDB dataset (Maas et al., 2011) from the `shapiq` benchmark. Lastly for the CIFAR10 game, we used a vision transformer (vit-base-patch16-224-in21k) (Dosovitskiy et al., 2021) fine-tuned on CIFAR10 (Krizhevsky et al., 2009), which is publicly available[3].

**Datasets.** The datasets and their source used for the tabular explanation games are described in Table 3. The *Forest Fires*[4] and *Real Estate*[5] were sourced from UCI Machine Learning Repository (UCI Repo), whereas *Bike Regression* was taken from OpenML (Feurer et al., 2020). The *California Housing* dataset was sourced from scikit-learn (Pedregosa et al., 2011)[`sklearn`], and the remaining datasets were sourced from the `shap`[6] library.

**Random forest configuration.** We use the standard implementation for `RandomForestRegressor` and `RandomForestClassifier` from `scikit-learn` (Pedregosa et al., 2011) with 10 tree instances of maximum depth 10 and fit the training data using accuracy (classification) and $R^2$ (regression). For all datasets, a $80/20$ percent train-test-split was executed.

**RegressionMSR.** For the RegressionMSR approach, we use `XGBoost` (Chen & Guestrin, 2016) with its default configuration as a tree-based backbone combined with the `MonteCarlo` approximator (equivalent to MSR (Witter et al., 2025)) from the `shapiq` package.

---

[2]`https://developer.nvidia.com/cuda-toolkit`
[3]`https://huggingface.co/aaraki/vit-base-patch16-224-in21k-finetuned-cifar10`
[4]`https://archive.ics.uci.edu/ml/datasets/forest+fires`
[5]`https://archive.ics.uci.edu/dataset/477/real+estate+valuation+data+set`
[6]`https://shap.readthedocs.io/en/latest/`

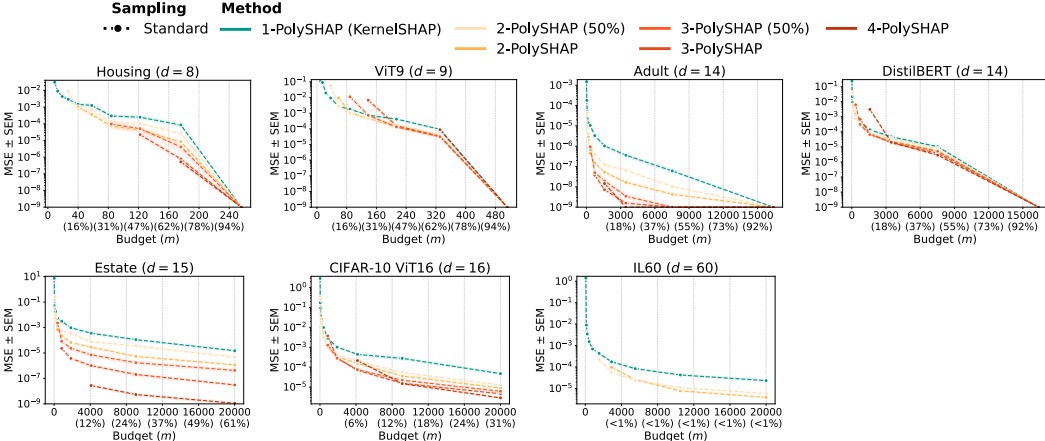

Figure 5: Approximation quality measured by MSE ($\pm$ SEM) for varying budget ($m$) on remaining explanation games. Adding interactions in PolySHAP can substantially improve approximation quality

**MSR and Unbiased KernelSHAP.** Fumagalli et al. (2023, Theorem 4.5) established that MSR (Wang & Jia, 2023) is equivalent to Unbiased KernelSHAP (Covert & Lee, 2021). We use the implementation of Unbiased KernelSHAP provided in the shapiq package.

**SVARM** Shapley Value Approximation without Requesting Marginals (SVARM) was proposed by Kolpaczki et al. (2024) and uses stratification of MSR (Castro et al., 2017). We use the implementation of SVARM provided in the shapiq package.

## B.2 ADDITIONAL RESULTS ON APPROXIMATION QUALITY USING MSE

In this section, we report approximation quality measured by MSE for the remaining explanation games.

Figure 5 reports the MSE for the *Housing*, *ViT9*, *Adult*, *DistilBERT*, *Estate*, and *IL60* explanation games. Similar to Figure 2, we observe that PolySHAP's approximation quality substantially improves with higher-order interactions. Again, this comes at the cost of larger budget requirements, indicated by the delay of the line plots. The Permutation Sampling and KernelSHAP (1-PolySHAP) baseline are consistently outperformed by higher-order PolySHAP, while RegressionMSR yields comparable results.

Figure 6 shows the approximation quality of PolySHAP with and without (standard) paired subset sampling. Similar to Figure 3, we observe a strong improvement of 1-PolySHAP due to the equivalence to 2-PolySHAP. The same observation holds for 3-PolySHAP.

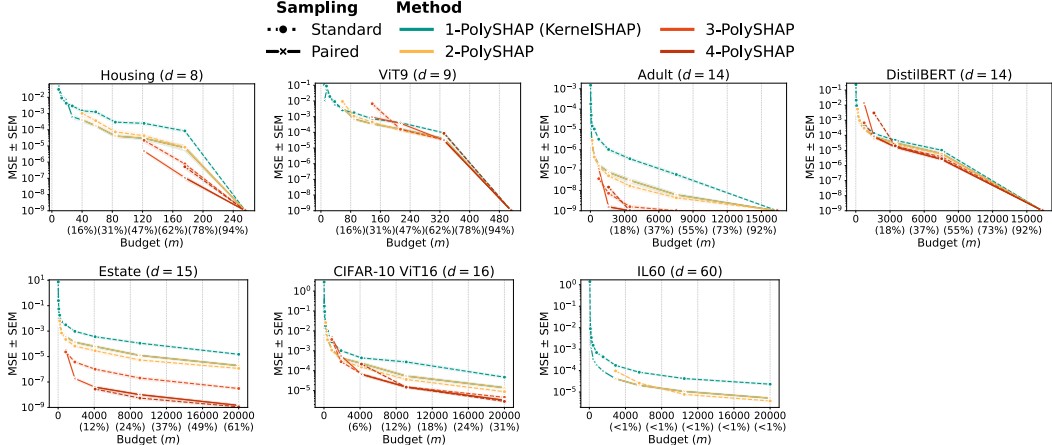

Figure 6: Approximation quality measured by MSE ($\pm$ SEM) for standard (dotted) and paired (solid) sampling for remaining local explanation games. Under paired sampling, 2-PolySHAP marginally improves, whereas KernelSHAP substantially improves due to its equivalence to 2-PolySHAP.

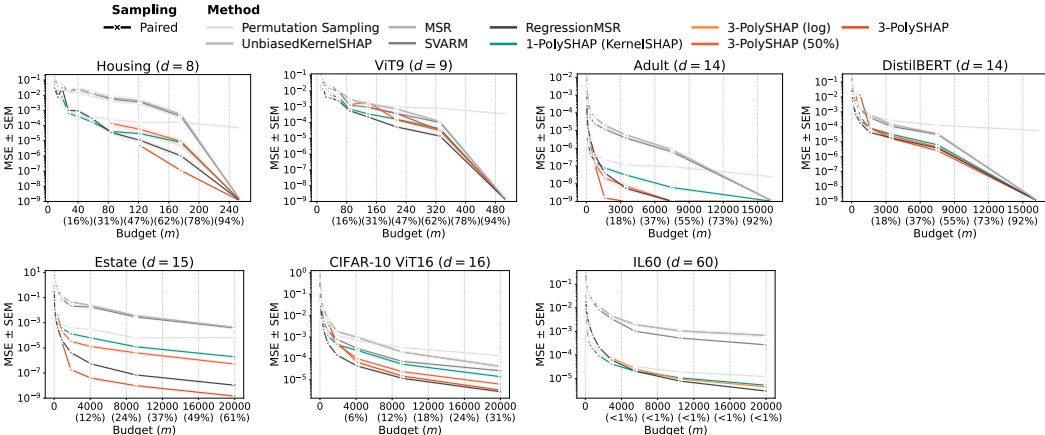

Figure 7: Approximation quality of PolySHAP variants and baselines measured by MSE ($\pm$ SEM) for paired sampling for remaining local explanation games.

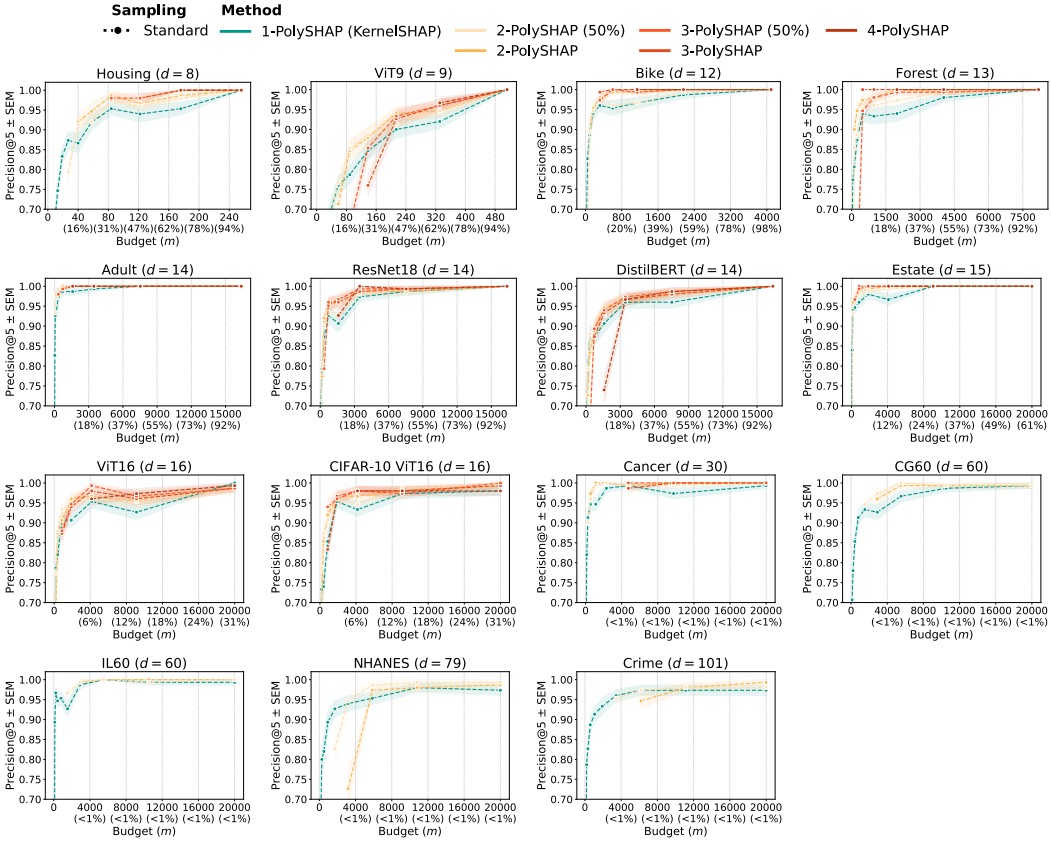

Figure 8: Approximation quality measured by Precision@5 ($\pm$ SEM) for varying budget ($m$) on different games. Adding interactions in PolySHAP can substantially improve approximation quality

### B.3 APPROXIMATION QUALITY USING PRECISION@5

In this section, we report approximation quality with respect to top-5 precision (*Precision@5*) for all explanation games from Table 2.

In Figure 8 we observe that higher-order interactions also improve the approximation quality regarding the Precision@5 metric. However, the distinction is not as clear as for MSE, since ranking is not considered in the optimization objective. In general, the approximation quality varies across different games, where the low-dimensional tabular explanation games show very good results, in contrast to the more challenging non-tabular games (ViT9, DistilBERT, ResNet18 and ViT16), and high-dimensional games (CG60, IL60, NHANES, and Crime), which require more budget for similar results.

In Figure 9 Precision@5 is compared for standard sampling and paired sampling. Again, we observe improvements for 1-PolySHAP and 3-PolySHAP when using paired sampling due to its equivalence to 2-PolySHAP and 4-PolySHAP, respectively.

In Figure 10, we report the Precision@5 metric for the PolySHAP variants and the baselines for paired sampling. Again, we observe state-of-the-art performance for PolySHAP and Regression-MSR. PolySHAP's performance substantially improves, if the budget allows to capture order-3 interactions.

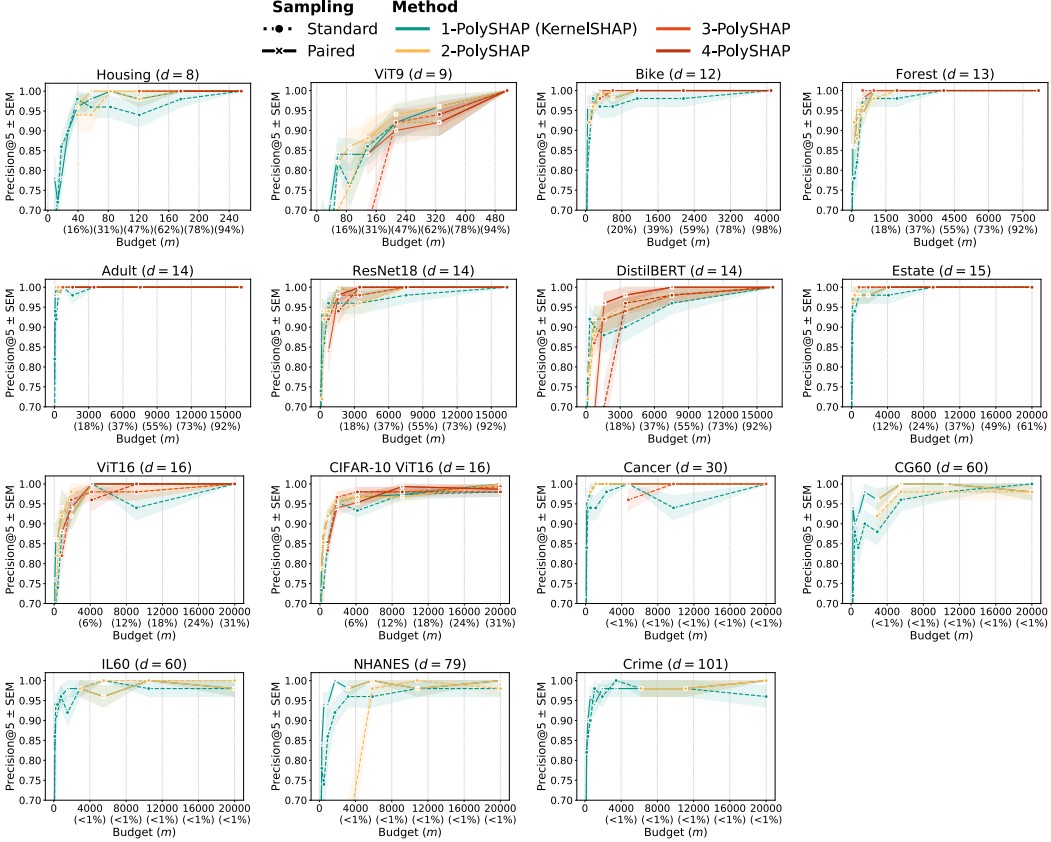

Figure 9: Approximation quality measured by Precision@5 ($\pm$ SEM) for standard (dotted) and paired (solid) sampling. Under paired sampling, 2-PolySHAP marginally improves, whereas KernelSHAP substantially improves due to its equivalence to 2-PolySHAP

## B.4 APPROXIMATION QUALITY USING SPEARMAN CORRELATION

In this section, we report approximation quality with respect to Spearman correlation (*Spearman-Correlation*) for all explanation games from Table 2.

Figure 11 reports Spearman correlation of PolySHAP and the baseline methods. Again, we observe consistent improvements of higher-order interactions in this metric. For high-dimensional settings ($\geq 60$), we further observe that the baselines clearly outperform PolySHAP in this metric. Since we have seen that PolySHAP performs very well in the Precision@5 metric, we conjecture that this difference is mainly due to features with lower absolute Shapley values.

In Figure 12, we observe a similar pattern as with MSE and Precision@5. Using paired sampling drastically improves the approximation quality of 1-PolySHAP, due to its equivalence to 2-PolySHAP. Since 3-PolySHAP often performs very well in this metric, we do not observe strong differences between 3-PolySHAP and 4-PolySHAP in both sampling settings.

In Figure 13, we report SpearmanCorrelation for the PolySHAP variants and the baseline methods under paired sampling. Again, we observe state-of-the-art performance for PolySHAP and the RegressionMSR baseline. PolySHAP substantially improves, if the budget allows to capture order-3 interactions.

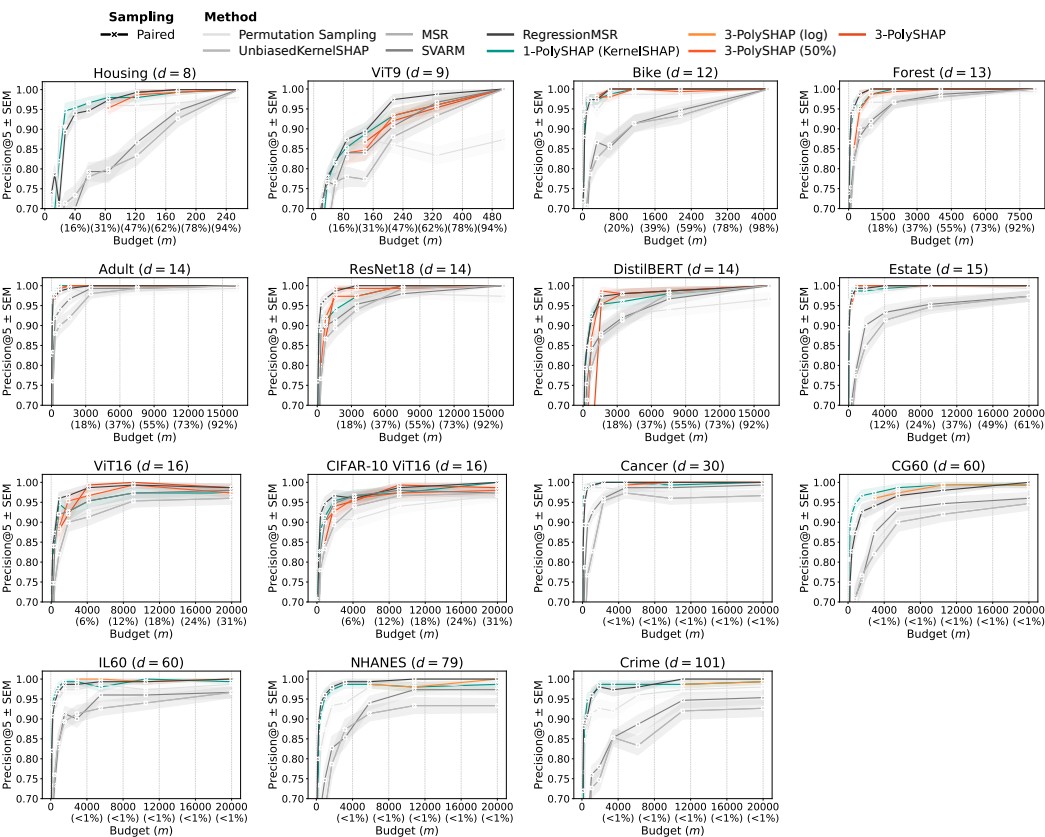

Figure 10: Approximation quality of PolySHAP variants and baselines measured by Precision@5 ($\pm$ SEM) for paired sampling

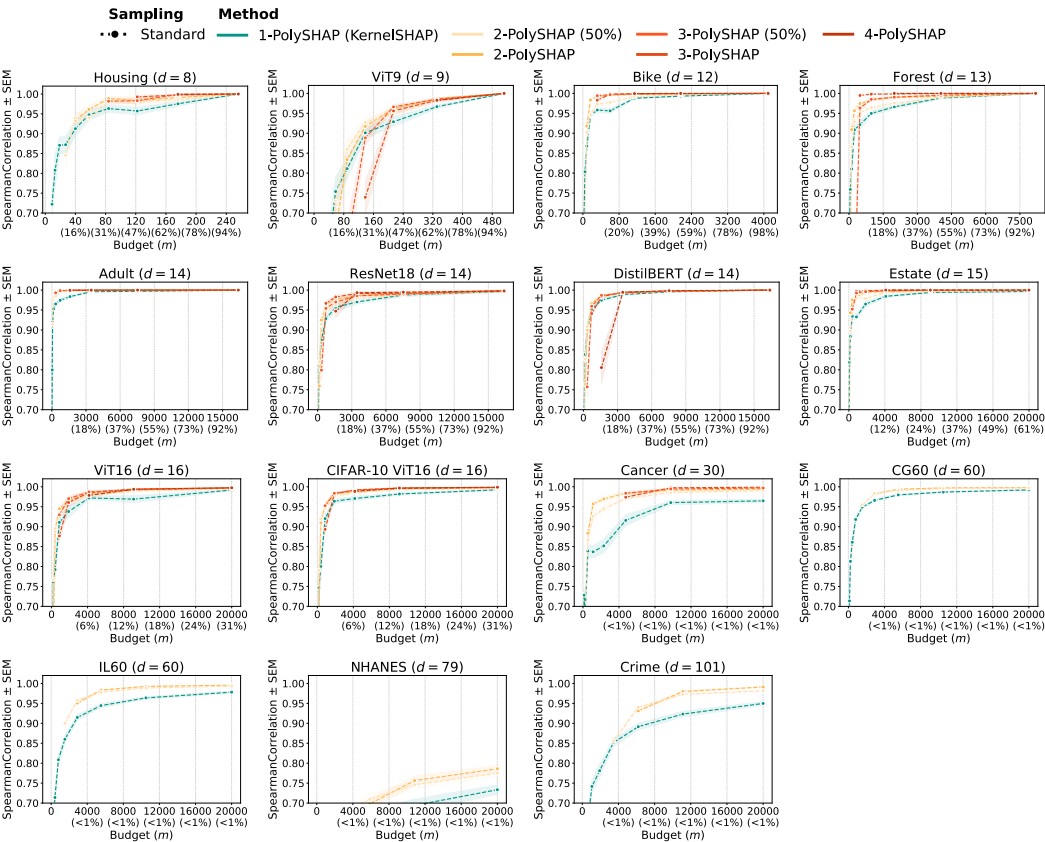

Figure 11: Approximation quality measured by SpearmanCorrelation ($\pm$ SEM) for varying budget ($m$) on different games. Adding interactions in PolySHAP can substantially improve approximation quality

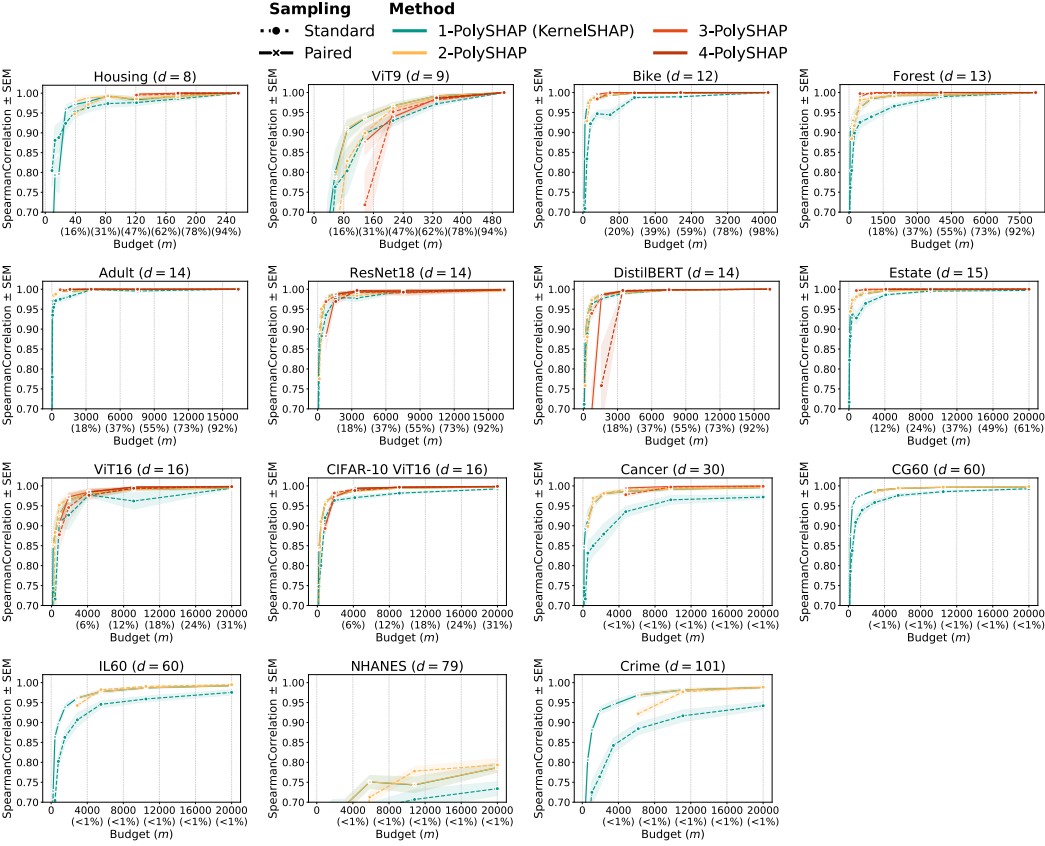

Figure 12: Approximation quality measured by SpearmanCorrelation ($\pm$ SEM) for standard (dotted) and paired (solid) sampling. Under paired sampling, 2-PolySHAP marginally improves, whereas KernelSHAP substantially improves due to its equivalence to 2-PolySHAP

## B.5 RUNTIME ANALYSIS

In this section, we analyze the runtime of PolySHAP and the RegressionMSR baseline, since both methods approximate the game values, and subsequently extract Shapley value estimates.

Figure 14 reports the runtime in seconds (log-scale) of PolySHAP and RegressionMSR for the spent budget on different explanation games. As expected, we observe a *linear* relationship between the budget $m$ and the computation time in PolySHAP, indicated by the overlapping linear fits (dashed lines). Overall, the computational overhead of the computations executed in RegressionMSR and PolySHAP variants after game evaluations will be negligible in most application settings.

**Complexity of Evaluations.**   In realistic application settings, the runtime for game evaluations should be considered a main driver of computational complexity of PolySHAP and RegressionMSR. This is verified by the CIFAR10 ViT16 game in Figure 14, a), which requires one model call of the ViT16 for each game evaluation. The computational difference between RegressionMSR and all PolySHAP variants are thereby negligible.

For the runtime of the path-dependent tree games, reported in Figure 14, b) the game evaluations require only a single pass through the random forests, which becomes negligible with increasing dimensionality.

**Complexity of Computation.**   The computational overhead of RegressionMSR and PolySHAP variants besides the game evaluations is negligible in many application settings. However, there is an impact on runtime for the higher-order $k$-PolySHAP variants, due to the increasing number

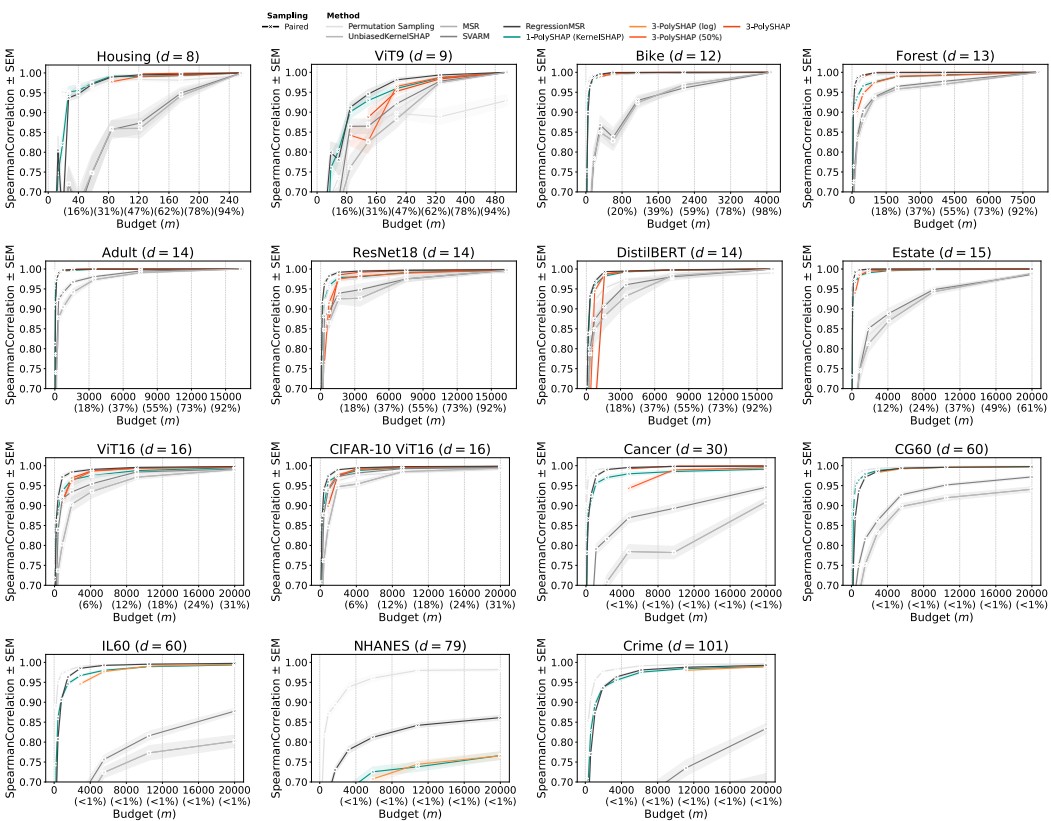

Figure 13: Approximation quality of PolySHAP variants and baselines measured by SpearmanCorrelation (± SEM) for paired sampling

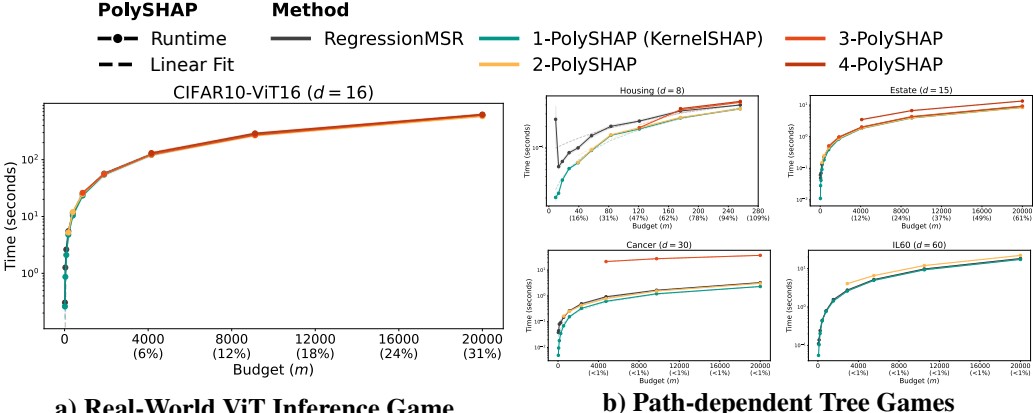

Figure 14: Runtime in seconds (log-scale) of PolySHAP and RegressionMSR for varying budgets ($m$) of **a)** a real-world ViT inference game on CIFAR10, and **b)** selected path-dependent tree games of varying dimensionality. The runtime increases linearly with the budget $m$, indicated by the linear fit (dashed line). As expected in practice, the runtime of the real-world ViT inference game on CIFAR10 is dominated by the model calls required for each budget. The computational differences of the approximators are negligible. For path-dependent tree games that require only a single tree traversal per game evaluation, the runtime increases for higher-order PolySHAP due to the increasing number of regression variables with stronger effects in high-dimensional games.

of regression variables that yield a *polynomial* increase of computation time. For the tree-path dependent games in Figure 14, b) this effect is visible due to the very efficient computation of game values.

The RegressionMSR method utilizes the XGBoost library (Chen & Guestrin, 2016), which scales well to high-dimensional problems, indicated by the low runtime observed in Figure 14. The runtime of these computations is generally higher than 1- and 2-PolySHAP, but less than 3- and 4-PolySHAP for high-dimensional problems.

## B.6 ADDITIONAL TABLES

Table 4: Summary statistics of the MSE error for ALL Shapley value estimators we consider with paired sampling. Increasing the degree of PolySHAP improves its performance, but $k$-PolySHAP requires budget $m \geq d_k = \mathcal{O}(k)$. RegressionMSR with XGBoost performs very well, except on games like CG60 or Crime where the decision tree struggles to approximate $\nu$.

| | Housing ($d=8$) | ViT9 ($d=9$) | Bike ($d=12$) | Forest ($d=13$) | Adult ($d=14$) | ResNet18 ($d=14$) | DistilBERT ($d=14$) | Estate ($d=15$) | ViT16 ($d=16$) | Cancer ($d=30$) | IL60 ($d=60$) | CG60 ($d=60$) | NHANES ($d=79$) | Crime ($d=101$) |
|---|---|---|---|---|---|---|---|---|---|---|---|---|---|---|
| $m$ | | | 1140 | 1988 | | 1590 | | | 4156 | 4749 | | 2900 | 3174 | 6188 |
| **Permutation Sampling** | | | | | | | | | | | | | | |
| Mean | $1.7 \times 10^{-4}$ | $7.6 \times 10^{-4}$ | $7.4 \times 10^{-1}$ | $1.6 \times 10^{-2}$ | $3.3 \times 10^{-7}$ | $1.0 \times 10^{-4}$ | $5.7 \times 10^{-4}$ | $2.0 \times 10^{-4}$ | $3.7 \times 10^{-5}$ | $7.2 \times 10^{-7}$ | $7.5 \times 10^{-5}$ | $6.0 \times 10^{-5}$ | $3.1 \times 10^{-3}$ | $6.4 \times 10^{-1}$ |
| 1st Quartile | $3.2 \times 10^{-5}$ | $2.9 \times 10^{-4}$ | $2.1 \times 10^{-1}$ | $4.6 \times 10^{-3}$ | $2.0 \times 10^{-7}$ | $2.1 \times 10^{-5}$ | $2.3 \times 10^{-4}$ | $3.5 \times 10^{-5}$ | $1.9 \times 10^{-5}$ | $4.2 \times 10^{-7}$ | $5.1 \times 10^{-5}$ | $5.0 \times 10^{-5}$ | $1.1 \times 10^{-3}$ | $1.8 \times 10^{-1}$ |
| 2nd Quartile | $1.5 \times 10^{-4}$ | $5.7 \times 10^{-4}$ | $5.1 \times 10^{-1}$ | $1.3 \times 10^{-2}$ | $2.4 \times 10^{-7}$ | $3.6 \times 10^{-5}$ | $5.8 \times 10^{-4}$ | $8.6 \times 10^{-5}$ | $3.2 \times 10^{-5}$ | $7.0 \times 10^{-7}$ | $7.1 \times 10^{-5}$ | $5.7 \times 10^{-5}$ | $4.0 \times 10^{-3}$ | $3.4 \times 10^{-1}$ |
| 3rd Quartile | $2.2 \times 10^{-4}$ | $1.0 \times 10^{-3}$ | $7.7 \times 10^{-1}$ | $1.6 \times 10^{-2}$ | $4.1 \times 10^{-7}$ | $2.1 \times 10^{-4}$ | $9.5 \times 10^{-4}$ | $3.5 \times 10^{-5}$ | $8.5 \times 10^{-7}$ | $9.8 \times 10^{-5}$ | $7.6 \times 10^{-5}$ | $4.4 \times 10^{-3}$ | $6.7 \times 10^{-1}$ |
| **1-PolySHAP (KernelSHAP)** | | | | | | | | | | | | | | |
| Mean | $5.5 \times 10^{-6}$ | $3.3 \times 10^{-5}$ | $4.0 \times 10^{-2}$ | $4.9 \times 10^{-3}$ | $1.2 \times 10^{-7}$ | $1.9 \times 10^{-5}$ | $1.0 \times 10^{-4}$ | $5.2 \times 10^{-5}$ | $1.5 \times 10^{-5}$ | $5.5 \times 10^{-7}$ | $4.7 \times 10^{-5}$ | $4.3 \times 10^{-5}$ | $2.6 \times 10^{-3}$ | $5.7 \times 10^{-1}$ |
| 1st Quartile | $1.3 \times 10^{-6}$ | $6.7 \times 10^{-6}$ | $1.7 \times 10^{-2}$ | $4.5 \times 10^{-4}$ | $4.6 \times 10^{-8}$ | $5.5 \times 10^{-6}$ | $3.8 \times 10^{-5}$ | $2.2 \times 10^{-5}$ | $7.5 \times 10^{-6}$ | $9.0 \times 10^{-8}$ | $3.6 \times 10^{-5}$ | $3.0 \times 10^{-5}$ | $1.1 \times 10^{-3}$ | $1.1 \times 10^{-1}$ |
| 2nd Quartile | $4.3 \times 10^{-6}$ | $1.1 \times 10^{-5}$ | $2.5 \times 10^{-2}$ | $1.2 \times 10^{-3}$ | $9.3 \times 10^{-8}$ | $1.3 \times 10^{-5}$ | $1.1 \times 10^{-4}$ | $4.0 \times 10^{-5}$ | $1.5 \times 10^{-5}$ | $2.1 \times 10^{-7}$ | $4.0 \times 10^{-5}$ | $3.7 \times 10^{-5}$ | $2.3 \times 10^{-3}$ | $1.9 \times 10^{-1}$ |
| 3rd Quartile | $5.8 \times 10^{-6}$ | $6.6 \times 10^{-5}$ | $4.8 \times 10^{-2}$ | $8.1 \times 10^{-3}$ | $1.1 \times 10^{-7}$ | $2.9 \times 10^{-5}$ | $1.3 \times 10^{-4}$ | $5.8 \times 10^{-5}$ | $2.2 \times 10^{-5}$ | $7.6 \times 10^{-7}$ | $5.3 \times 10^{-5}$ | $4.2 \times 10^{-5}$ | $4.2 \times 10^{-3}$ | $6.1 \times 10^{-1}$ |
| **2-PolySHAP (50%)** | | | | | | | | | | | | | | |
| Mean | $5.5 \times 10^{-6}$ | $3.3 \times 10^{-5}$ | $4.0 \times 10^{-2}$ | $4.9 \times 10^{-3}$ | $1.2 \times 10^{-7}$ | $1.9 \times 10^{-5}$ | $1.0 \times 10^{-4}$ | $5.2 \times 10^{-5}$ | $1.5 \times 10^{-5}$ | $5.4 \times 10^{-7}$ | $4.7 \times 10^{-5}$ | $4.3 \times 10^{-5}$ | $2.6 \times 10^{-3}$ | $5.7 \times 10^{-1}$ |
| 1st Quartile | $1.3 \times 10^{-6}$ | $6.7 \times 10^{-6}$ | $1.7 \times 10^{-2}$ | $4.5 \times 10^{-4}$ | $4.6 \times 10^{-8}$ | $5.5 \times 10^{-6}$ | $3.8 \times 10^{-5}$ | $2.2 \times 10^{-5}$ | $7.5 \times 10^{-6}$ | $9.2 \times 10^{-8}$ | $3.6 \times 10^{-5}$ | $3.0 \times 10^{-5}$ | $1.1 \times 10^{-3}$ | $1.1 \times 10^{-1}$ |
| 2nd Quartile | $4.3 \times 10^{-6}$ | $1.1 \times 10^{-5}$ | $2.5 \times 10^{-2}$ | $1.2 \times 10^{-3}$ | $9.3 \times 10^{-8}$ | $1.3 \times 10^{-5}$ | $1.1 \times 10^{-4}$ | $4.0 \times 10^{-5}$ | $1.5 \times 10^{-5}$ | $2.1 \times 10^{-7}$ | $4.0 \times 10^{-5}$ | $3.7 \times 10^{-5}$ | $2.3 \times 10^{-3}$ | $1.9 \times 10^{-1}$ |
| 3rd Quartile | $5.8 \times 10^{-6}$ | $6.6 \times 10^{-5}$ | $4.8 \times 10^{-2}$ | $8.1 \times 10^{-3}$ | $1.1 \times 10^{-7}$ | $2.9 \times 10^{-5}$ | $1.3 \times 10^{-4}$ | $5.8 \times 10^{-5}$ | $2.2 \times 10^{-5}$ | $7.6 \times 10^{-7}$ | $5.3 \times 10^{-5}$ | $4.2 \times 10^{-5}$ | $4.2 \times 10^{-3}$ | $6.1 \times 10^{-1}$ |
| **2-PolySHAP** | | | | | | | | | | | | | | |
| Mean | $5.5 \times 10^{-6}$ | $3.3 \times 10^{-5}$ | $4.0 \times 10^{-2}$ | $4.9 \times 10^{-3}$ | $1.2 \times 10^{-7}$ | $1.9 \times 10^{-5}$ | $1.0 \times 10^{-4}$ | $5.2 \times 10^{-5}$ | $1.5 \times 10^{-5}$ | $5.4 \times 10^{-7}$ | $4.7 \times 10^{-5}$ | $4.3 \times 10^{-5}$ | $2.6 \times 10^{-3}$ | $5.7 \times 10^{-1}$ |
| 1st Quartile | $1.3 \times 10^{-6}$ | $6.7 \times 10^{-6}$ | $1.7 \times 10^{-2}$ | $4.5 \times 10^{-4}$ | $4.6 \times 10^{-8}$ | $5.5 \times 10^{-6}$ | $3.8 \times 10^{-5}$ | $2.2 \times 10^{-5}$ | $7.5 \times 10^{-6}$ | $9.2 \times 10^{-8}$ | $3.6 \times 10^{-5}$ | $3.0 \times 10^{-5}$ | $1.1 \times 10^{-3}$ | $1.1 \times 10^{-1}$ |
| 2nd Quartile | $4.3 \times 10^{-6}$ | $1.1 \times 10^{-5}$ | $2.5 \times 10^{-2}$ | $1.2 \times 10^{-3}$ | $9.3 \times 10^{-8}$ | $1.3 \times 10^{-5}$ | $1.1 \times 10^{-4}$ | $4.0 \times 10^{-5}$ | $1.5 \times 10^{-5}$ | $2.1 \times 10^{-7}$ | $4.0 \times 10^{-5}$ | $3.7 \times 10^{-5}$ | $2.3 \times 10^{-3}$ | $1.9 \times 10^{-1}$ |
| 3rd Quartile | $5.8 \times 10^{-6}$ | $6.6 \times 10^{-5}$ | $4.8 \times 10^{-2}$ | $8.1 \times 10^{-3}$ | $1.1 \times 10^{-7}$ | $2.9 \times 10^{-5}$ | $1.3 \times 10^{-4}$ | $5.8 \times 10^{-5}$ | $2.2 \times 10^{-5}$ | $7.6 \times 10^{-7}$ | $5.3 \times 10^{-5}$ | $4.2 \times 10^{-5}$ | $4.2 \times 10^{-3}$ | $6.1 \times 10^{-1}$ |
| **3-PolySHAP (50%)** | | | | | | | | | | | | | | |
| Mean | $4.1 \times 10^{-6}$ | $2.7 \times 10^{-5}$ | $3.5 \times 10^{-2}$ | $1.7 \times 10^{-3}$ | $6.5 \times 10^{-8}$ | $9.1 \times 10^{-6}$ | $5.3 \times 10^{-5}$ | $1.0 \times 10^{-5}$ | $7.9 \times 10^{-6}$ | $2.0 \times 10^{-6}$ | | | | |
| 1st Quartile | $6.5 \times 10^{-7}$ | $8.2 \times 10^{-6}$ | $7.6 \times 10^{-3}$ | $9.8 \times 10^{-5}$ | $2.5 \times 10^{-8}$ | $2.4 \times 10^{-6}$ | $2.9 \times 10^{-5}$ | $5.1 \times 10^{-6}$ | $3.3 \times 10^{-6}$ | $4.8 \times 10^{-7}$ | | | | |
| 2nd Quartile | $1.4 \times 10^{-6}$ | $1.9 \times 10^{-5}$ | $1.4 \times 10^{-2}$ | $2.1 \times 10^{-4}$ | $3.9 \times 10^{-8}$ | $4.0 \times 10^{-6}$ | $4.2 \times 10^{-5}$ | $8.5 \times 10^{-6}$ | $4.9 \times 10^{-6}$ | $8.5 \times 10^{-7}$ | | | | |
| 3rd Quartile | $3.4 \times 10^{-6}$ | $3.6 \times 10^{-5}$ | $3.5 \times 10^{-2}$ | $7.2 \times 10^{-4}$ | $8.8 \times 10^{-8}$ | $1.5 \times 10^{-5}$ | | $6.8 \times 10^{-6}$ | $1.1 \times 10^{-5}$ | $9.6 \times 10^{-6}$ | $1.9 \times 10^{-6}$ | | | |
| **3-PolySHAP** | | | | | | | | | | | | | | |
| Mean | $4.0 \times 10^{-8}$ | $2.7 \times 10^{-5}$ | $8.0 \times 10^{-4}$ | $4.3 \times 10^{-7}$ | $1.9 \times 10^{-9}$ | $6.9 \times 10^{-8}$ | $7.5 \times 10^{-5}$ | $2.7 \times 10^{-4}$ | $5.7 \times 10^{-6}$ | $3.2 \times 10^{-7}$ | | | | |
| 1st Quartile | $5.5 \times 10^{-9}$ | $5.6 \times 10^{-6}$ | $1.2 \times 10^{-4}$ | $1.7 \times 10^{-7}$ | $4.2 \times 10^{-10}$ | $1.1 \times 10^{-6}$ | $4.2 \times 10^{-5}$ | $1.4 \times 10^{-4}$ | $2.1 \times 10^{-6}$ | $1.2 \times 10^{-7}$ | | | | |
| 2nd Quartile | $1.5 \times 10^{-8}$ | $1.8 \times 10^{-5}$ | $1.8 \times 10^{-4}$ | $3.7 \times 10^{-7}$ | $1.3 \times 10^{-9}$ | $2.5 \times 10^{-6}$ | $6.6 \times 10^{-5}$ | $2.1 \times 10^{-4}$ | $3.0 \times 10^{-6}$ | $2.1 \times 10^{-7}$ | | | | |
| 3rd Quartile | $6.1 \times 10^{-8}$ | $4.0 \times 10^{-5}$ | $8.5 \times 10^{-4}$ | $7.1 \times 10^{-7}$ | $2.2 \times 10^{-9}$ | $8.2 \times 10^{-6}$ | $1.1 \times 10^{-4}$ | $3.0 \times 10^{-4}$ | $6.5 \times 10^{-6}$ | | | | | |
| **4-PolySHAP** | | | | | | | | | | | | | | |
| Mean | $4.0 \times 10^{-8}$ | $2.7 \times 10^{-5}$ | $8.0 \times 10^{-4}$ | $4.3 \times 10^{-7}$ | $1.9 \times 10^{-9}$ | $6.9 \times 10^{-8}$ | $7.5 \times 10^{-5}$ | $2.7 \times 10^{-4}$ | $5.7 \times 10^{-6}$ | | | | | |
| 1st Quartile | $5.5 \times 10^{-9}$ | $5.6 \times 10^{-6}$ | $1.2 \times 10^{-4}$ | $1.7 \times 10^{-7}$ | $4.2 \times 10^{-10}$ | $1.1 \times 10^{-6}$ | $4.2 \times 10^{-5}$ | $1.4 \times 10^{-4}$ | $2.1 \times 10^{-6}$ | | | | | |
| 2nd Quartile | $1.5 \times 10^{-8}$ | $1.8 \times 10^{-5}$ | $1.8 \times 10^{-4}$ | $3.7 \times 10^{-7}$ | $1.3 \times 10^{-9}$ | $2.5 \times 10^{-6}$ | $6.6 \times 10^{-5}$ | $2.1 \times 10^{-4}$ | $3.0 \times 10^{-6}$ | | | | | |
| 3rd Quartile | $6.1 \times 10^{-8}$ | $4.0 \times 10^{-5}$ | $8.5 \times 10^{-4}$ | $7.1 \times 10^{-7}$ | $2.2 \times 10^{-9}$ | $8.2 \times 10^{-6}$ | $1.1 \times 10^{-4}$ | $3.0 \times 10^{-4}$ | $6.5 \times 10^{-6}$ | | | | | |
| **RegressionMSR** | | | | | | | | | | | | | | |
| Mean | $4.3 \times 10^{-7}$ | $1.2 \times 10^{-5}$ | $2.0 \times 10^{-3}$ | $1.3 \times 10^{-5}$ | $6.0 \times 10^{-8}$ | $2.1 \times 10^{-6}$ | $4.1 \times 10^{-5}$ | $4.5 \times 10^{-7}$ | $3.4 \times 10^{-6}$ | $9.0 \times 10^{-8}$ | $5.8 \times 10^{-5}$ | $2.5 \times 10^{-4}$ | $5.3 \times 10^{-4}$ | $6.0 \times 10^{-1}$ |
| 1st Quartile | $9.3 \times 10^{-8}$ | $4.2 \times 10^{-6}$ | $6.4 \times 10^{-4}$ | $9.4 \times 10^{-6}$ | $3.6 \times 10^{-8}$ | $2.2 \times 10^{-7}$ | $1.8 \times 10^{-5}$ | $1.9 \times 10^{-7}$ | $6.8 \times 10^{-7}$ | $4.8 \times 10^{-8}$ | $2.0 \times 10^{-4}$ | $2.0 \times 10^{-4}$ | $2.4 \times 10^{-1}$ |
| 2nd Quartile | $1.4 \times 10^{-7}$ | $8.8 \times 10^{-6}$ | $1.2 \times 10^{-3}$ | $1.1 \times 10^{-5}$ | $5.2 \times 10^{-8}$ | $1.9 \times 10^{-6}$ | $3.3 \times 10^{-5}$ | $4.7 \times 10^{-7}$ | $1.9 \times 10^{-6}$ | $8.7 \times 10^{-8}$ | $6.5 \times 10^{-5}$ | $2.5 \times 10^{-4}$ | $4.2 \times 10^{-4}$ | $3.3 \times 10^{-1}$ |
| 3rd Quartile | $3.2 \times 10^{-7}$ | $1.8 \times 10^{-5}$ | $1.4 \times 10^{-3}$ | $1.5 \times 10^{-5}$ | $7.9 \times 10^{-8}$ | $3.0 \times 10^{-6}$ | $4.2 \times 10^{-5}$ | $6.6 \times 10^{-7}$ | $3.2 \times 10^{-6}$ | $1.2 \times 10^{-7}$ | $6.8 \times 10^{-5}$ | $2.6 \times 10^{-4}$ | $5.9 \times 10^{-4}$ | $6.7 \times 10^{-1}$ |

Table 5: Summary statistics of the MSE error for ALL Shapley value estimators we consider with standard (not paired) sampling. Increasing the degree of PolySHAP improves its performance, but $k$-PolySHAP requires budget $m \geq d_k = \mathcal{O}(k)$. RegressionMSR with XGBoost performs very well, except on games like CG60 or Crime where the decision tree struggles to approximate $\nu$.

| | Housing ($d=8$) | ViT9 ($d=9$) | Bike ($d=12$) | Forest ($d=13$) | Adult ($d=14$) | ResNet18 ($d=14$) | DistilBERT ($d=14$) | Estate ($d=15$) | ViT16 ($d=16$) | Cancer ($d=30$) | IL60 ($d=60$) | CG60 ($d=60$) | NHANES ($d=79$) | Crime ($d=101$) |
|---|---|---|---|---|---|---|---|---|---|---|---|---|---|---|
| $m$ | | | 1140 | 1988 | | 1590 | | | 4156 | 4749 | | 2900 | 3174 | 6188 |
| **Permutation Sampling** | | | | | | | | | | | | | | |
| Mean | $1.2 \times 10^{-3}$ | $1.2 \times 10^{-3}$ | $5.6 \times 10^{0}$ | $2.7 \times 10^{-2}$ | $2.6 \times 10^{-6}$ | $1.0 \times 10^{-4}$ | $6.3 \times 10^{-4}$ | $5.5 \times 10^{-4}$ | $6.0 \times 10^{-5}$ | $5.5 \times 10^{-6}$ | $1.0 \times 10^{-4}$ | $1.0 \times 10^{-4}$ | $7.6 \times 10^{-3}$ | $3.9 \times 10^{0}$ |
| 1st Quartile | $3.8 \times 10^{-4}$ | $3.1 \times 10^{-4}$ | $1.3 \times 10^{0}$ | $6.3 \times 10^{-3}$ | $6.9 \times 10^{-7}$ | $4.7 \times 10^{-5}$ | $3.1 \times 10^{-4}$ | $1.0 \times 10^{-4}$ | $1.6 \times 10^{-5}$ | $9.5 \times 10^{-7}$ | $7.7 \times 10^{-5}$ | $8.0 \times 10^{-5}$ | $2.6 \times 10^{-3}$ | $8.8 \times 10^{-1}$ |
| 2nd Quartile | $9.0 \times 10^{-4}$ | $9.9 \times 10^{-4}$ | $1.4 \times 10^{0}$ | $3.9 \times 10^{-2}$ | $1.5 \times 10^{-6}$ | $8.5 \times 10^{-5}$ | $5.4 \times 10^{-4}$ | $2.3 \times 10^{-4}$ | $4.7 \times 10^{-5}$ | $2.3 \times 10^{-6}$ | $9.7 \times 10^{-5}$ | $1.0 \times 10^{-4}$ | $5.2 \times 10^{-3}$ | $1.2 \times 10^{0}$ |
| 3rd Quartile | $1.4 \times 10^{-3}$ | $1.4 \times 10^{-3}$ | $3.2 \times 10^{0}$ | $3.9 \times 10^{-2}$ | $3.2 \times 10^{-6}$ | $1.6 \times 10^{-4}$ | $7.2 \times 10^{-4}$ | $1.1 \times 10^{-3}$ | $8.6 \times 10^{-5}$ | $3.7 \times 10^{-6}$ | $1.3 \times 10^{-4}$ | $1.2 \times 10^{-4}$ | $1.0 \times 10^{-2}$ | $1.4 \times 10^{0}$ |
| **1-PolySHAP (KernelSHAP)** | | | | | | | | | | | | | | |
| Mean | $4.2 \times 10^{-5}$ | $9.3 \times 10^{-5}$ | $8.4 \times 10^{-1}$ | $1.4 \times 10^{-2}$ | $1.3 \times 10^{-6}$ | $4.6 \times 10^{-5}$ | $1.5 \times 10^{-4}$ | $2.7 \times 10^{-4}$ | $2.9 \times 10^{-5}$ | $4.0 \times 10^{-6}$ | $1.8 \times 10^{-4}$ | $2.3 \times 10^{-4}$ | $1.2 \times 10^{-2}$ | $3.5 \times 10^{0}$ |
| 1st Quartile | $1.9 \times 10^{-5}$ | $2.7 \times 10^{-5}$ | $3.4 \times 10^{-1}$ | $8.1 \times 10^{-3}$ | $1.9 \times 10^{-7}$ | $1.0 \times 10^{-5}$ | $6.4 \times 10^{-5}$ | $1.4 \times 10^{-4}$ | $1.3 \times 10^{-5}$ | $1.4 \times 10^{-6}$ | $9.6 \times 10^{-5}$ | $1.1 \times 10^{-4}$ | $3.5 \times 10^{-3}$ | $8.6 \times 10^{-1}$ |
| 2nd Quartile | $3.9 \times 10^{-5}$ | $5.1 \times 10^{-5}$ | $5.6 \times 10^{-1}$ | $2.5 \times 10^{-2}$ | $7.1 \times 10^{-7}$ | $2.5 \times 10^{-5}$ | $1.2 \times 10^{-4}$ | $2.0 \times 10^{-4}$ | $2.7 \times 10^{-5}$ | $2.3 \times 10^{-6}$ | $1.4 \times 10^{-4}$ | $1.4 \times 10^{-4}$ | $7.6 \times 10^{-3}$ | $1.4 \times 10^{0}$ |
| 3rd Quartile | $6.2 \times 10^{-5}$ | $1.5 \times 10^{-4}$ | $9.7 \times 10^{-1}$ | $3.5 \times 10^{-2}$ | $1.6 \times 10^{-6}$ | $8.8 \times 10^{-5}$ | $1.8 \times 10^{-4}$ | $2.7 \times 10^{-4}$ | $4.8 \times 10^{-5}$ | $4.1 \times 10^{-6}$ | $2.3 \times 10^{-4}$ | $1.8 \times 10^{-4}$ | $2.2 \times 10^{-2}$ | $1.8 \times 10^{0}$ |
| **2-PolySHAP (50%)** | | | | | | | | | | | | | | |
| Mean | $2.2 \times 10^{-5}$ | $4.0 \times 10^{-5}$ | $2.1 \times 10^{-1}$ | $5.9 \times 10^{-3}$ | $3.0 \times 10^{-7}$ | $1.8 \times 10^{-5}$ | $8.0 \times 10^{-5}$ | $7.9 \times 10^{-5}$ | $1.2 \times 10^{-5}$ | $7.0 \times 10^{-7}$ | $7.3 \times 10^{-5}$ | $5.8 \times 10^{-5}$ | $5.0 \times 10^{-3}$ | $1.2 \times 10^{0}$ |
| 1st Quartile | $7.5 \times 10^{-6}$ | $1.0 \times 10^{-5}$ | $7.5 \times 10^{-2}$ | $1.4 \times 10^{-3}$ | $2.0 \times 10^{-7}$ | $6.7 \times 10^{-6}$ | $2.3 \times 10^{-5}$ | $1.6 \times 10^{-5}$ | $6.0 \times 10^{-6}$ | $2.3 \times 10^{-7}$ | $5.1 \times 10^{-5}$ | $4.1 \times 10^{-5}$ | $1.7 \times 10^{-3}$ | $2.5 \times 10^{-1}$ |
| 2nd Quartile | $1.6 \times 10^{-5}$ | $2.1 \times 10^{-5}$ | $1.3 \times 10^{-1}$ | $3.1 \times 10^{-3}$ | $3.1 \times 10^{-7}$ | $1.3 \times 10^{-5}$ | $4.9 \times 10^{-5}$ | $3.5 \times 10^{-5}$ | $1.0 \times 10^{-5}$ | $3.2 \times 10^{-7}$ | $6.7 \times 10^{-5}$ | $6.3 \times 10^{-5}$ | $3.9 \times 10^{-3}$ | $9.1 \times 10^{-1}$ |
| 3rd Quartile | $3.2 \times 10^{-5}$ | $3.5 \times 10^{-5}$ | $3.5 \times 10^{-1}$ | $7.0 \times 10^{-3}$ | $3.8 \times 10^{-7}$ | $2.6 \times 10^{-5}$ | $9.0 \times 10^{-5}$ | $1.1 \times 10^{-4}$ | $1.7 \times 10^{-5}$ | $1.0 \times 10^{-6}$ | $8.4 \times 10^{-5}$ | $7.4 \times 10^{-5}$ | $9.0 \times 10^{-3}$ | $9.1 \times 10^{-1}$ |
| **2-PolySHAP** | | | | | | | | | | | | | | |
| Mean | $1.9 \times 10^{-5}$ | $3.4 \times 10^{-5}$ | $3.6 \times 10^{-2}$ | $1.5 \times 10^{-3}$ | $7.6 \times 10^{-8}$ | $1.4 \times 10^{-5}$ | $5.3 \times 10^{-5}$ | $2.7 \times 10^{-5}$ | $9.7 \times 10^{-6}$ | $2.9 \times 10^{-7}$ | $1.1 \times 10^{-4}$ | $9.5 \times 10^{-5}$ | $2.4 \times 10^{-1}$ | $3.5 \times 10^{0}$ |
| 1st Quartile | $2.3 \times 10^{-6}$ | $1.3 \times 10^{-5}$ | $1.7 \times 10^{-2}$ | $4.0 \times 10^{-4}$ | $2.5 \times 10^{-8}$ | $1.5 \times 10^{-5}$ | $1.3 \times 10^{-5}$ | $4.6 \times 10^{-6}$ | $6.8 \times 10^{-8}$ | $7.8 \times 10^{-5}$ | $6.1 \times 10^{-5}$ | $9.7 \times 10^{-2}$ | $5.7 \times 10^{-1}$ |
| 2nd Quartile | $7.6 \times 10^{-6}$ | $2.0 \times 10^{-5}$ | $3.9 \times 10^{-2}$ | $1.2 \times 10^{-3}$ | $5.9 \times 10^{-8}$ | $1.0 \times 10^{-5}$ | $3.8 \times 10^{-5}$ | $2.6 \times 10^{-5}$ | $8.7 \times 10^{-6}$ | $1.5 \times 10^{-7}$ | $9.8 \times 10^{-5}$ | $9.2 \times 10^{-5}$ | $2.0 \times 10^{-1}$ | $1.2 \times 10^{0}$ |
| 3rd Quartile | $1.6 \times 10^{-5}$ | $4.0 \times 10^{-5}$ | $4.8 \times 10^{-2}$ | $1.6 \times 10^{-3}$ | $2.0 \times 10^{-5}$ | $9.1 \times 10^{-5}$ | $3.2 \times 10^{-5}$ | $1.5 \times 10^{-4}$ | $1.1 \times 10^{-4}$ | $2.5 \times 10^{-1}$ | $3.9 \times 10^{0}$ |
| **3-PolySHAP (50%)** | | | | | | | | | | | | | | |
| Mean | $2.7 \times 10^{-5}$ | $3.9 \times 10^{-5}$ | $1.9 \times 10^{-2}$ | $6.6 \times 10^{-4}$ | $4.4 \times 10^{-8}$ | $1.3 \times 10^{-5}$ | $5.5 \times 10^{-5}$ | $5.8 \times 10^{-6}$ | $7.9 \times 10^{-6}$ | $2.3 \times 10^{-7}$ | | | | |
| 1st Quartile | $1.1 \times 10^{-6}$ | $5.9 \times 10^{-6}$ | $2.7 \times 10^{-3}$ | $6.6 \times 10^{-5}$ | $2.2 \times 10^{-8}$ | $1.4 \times 10^{-6}$ | $1.4 \times 10^{-5}$ | $1.1 \times 10^{-6}$ | $2.7 \times 10^{-6}$ | $6.6 \times 10^{-8}$ | | | | |
| 2nd Quartile | $2.3 \times 10^{-6}$ | $2.1 \times 10^{-5}$ | $8.1 \times 10^{-3}$ | $2.9 \times 10^{-4}$ | $3.6 \times 10^{-8}$ | $3.2 \times 10^{-6}$ | $3.8 \times 10^{-5}$ | $1.9 \times 10^{-6}$ | $4.5 \times 10^{-6}$ | $1.1 \times 10^{-7}$ | | | | |
| 3rd Quartile | $2.5 \times 10^{-5}$ | $4.5 \times 10^{-5}$ | $3.8 \times 10^{-2}$ | $6.4 \times 10^{-4}$ | $6.4 \times 10^{-8}$ | $1.7 \times 10^{-5}$ | $6.3 \times 10^{-5}$ | $8.5 \times 10^{-6}$ | $1.4 \times 10^{-5}$ | $2.4 \times 10^{-7}$ | | | | |
| **3-PolySHAP** | | | | | | | | | | | | | | |
| Mean | $9.7 \times 10^{-7}$ | $2.3 \times 10^{-5}$ | $4.8 \times 10^{-3}$ | $1.8 \times 10^{-5}$ | $9.1 \times 10^{-8}$ | $1.3 \times 10^{-5}$ | $6.5 \times 10^{-5}$ | $1.1 \times 10^{-5}$ | $4.6 \times 10^{-5}$ | $6.5 \times 10^{-6}$ | $2.0 \times 10^{-6}$ | | | |
| 1st Quartile | $1.1 \times 10^{-7}$ | $1.1 \times 10^{-5}$ | $1.0 \times 10^{-3}$ | $3.2 \times 10^{-6}$ | $3.4 \times 10^{-8}$ | $1.2 \times 10^{-6}$ | $3.3 \times 10^{-5}$ | $4.7 \times 10^{-7}$ | $2.0 \times 10^{-6}$ | $2.1 \times 10^{-7}$ | | | | |
| 2nd Quartile | $2.4 \times 10^{-7}$ | $1.2 \times 10^{-5}$ | $1.4 \times 10^{-3}$ | $6.9 \times 10^{-6}$ | $6.0 \times 10^{-8}$ | $5.2 \times 10^{-6}$ | $5.2 \times 10^{-5}$ | $1.3 \times 10^{-6}$ | $3.6 \times 10^{-6}$ | | | | | |
| 3rd Quartile | $8.0 \times 10^{-7}$ | $2.9 \times 10^{-5}$ | $3.8 \times 10^{-3}$ | $1.7 \times 10^{-5}$ | $1.3 \times 10^{-8}$ | $1.4 \times 10^{-5}$ | $8.6 \times 10^{-5}$ | $1.3 \times 10^{-6}$ | $1.0 \times 10^{-5}$ | $6.4 \times 10^{-7}$ | | | | |
| **4-PolySHAP** | | | | | | | | | | | | | | |
| Mean | $8.3 \times 10^{-7}$ | $6.7 \times 10^{-5}$ | $1.6 \times 10^{-3}$ | $1.2 \times 10^{-6}$ | $2.3 \times 10^{-8}$ | $6.1 \times 10^{-5}$ | $3.5 \times 10^{-3}$ | $2.0 \times 10^{-5}$ | $1.2 \times 10^{-5}$ | | | | | |
| 1st Quartile | $2.3 \times 10^{-8}$ | $8.4 \times 10^{-5}$ | $4.7 \times 10^{-4}$ | $1.6 \times 10^{-7}$ | $3.8 \times 10^{-9}$ | $9.1 \times 10^{-6}$ | $1.4 \times 10^{-3}$ | $8.8 \times 10^{-6}$ | $5.0 \times 10^{-6}$ | | | | | |
| 2nd Quartile | $8.4 \times 10^{-8}$ | $3.2 \times 10^{-5}$ | $6.0 \times 10^{-4}$ | $5.4 \times 10^{-7}$ | $1.6 \times 10^{-8}$ | $2.7 \times 10^{-5}$ | $2.4 \times 10^{-3}$ | $1.3 \times 10^{-5}$ | $9.3 \times 10^{-6}$ | | | | | |
| 3rd Quartile | $5.8 \times 10^{-7}$ | $7.2 \times 10^{-5}$ | $5.8 \times 10^{-4}$ | $9.1 \times 10^{-7}$ | $3.0 \times 10^{-8}$ | $7.7 \times 10^{-5}$ | $3.9 \times 10^{-3}$ | $2.3 \times 10^{-5}$ | $1.6 \times 10^{-5}$ | | | | | |
| **RegressionMSR** | | | | | | | | | | | | | | |
| Mean | $8.6 \times 10^{-7}$ | $1.3 \times 10^{-5}$ | $2.6 \times 10^{-3}$ | $1.5 \times 10^{-5}$ | $7.0 \times 10^{-8}$ | $3.5 \times 10^{-6}$ | $6.5 \times 10^{-5}$ | $4.6 \times 10^{-7}$ | $1.2 \times 10^{-5}$ | $7.3 \times 10^{-8}$ | $6.5 \times 10^{-5}$ | $2.4 \times 10^{-4}$ | $4.5 \times 10^{-4}$ | $5.8 \times 10^{-1}$ |
| 1st Quartile | $4.0 \times 10^{-7}$ | $8.7 \times 10^{-6}$ | $6.9 \times 10^{-6}$ | $5.6 \times 10^{-6}$ | $2.7 \times 10^{-7}$ | $2.5 \times 10^{-6}$ | $2.9 \times 10^{-7}$ | $1.6 \times 10^{-6}$ | $5.3 \times 10^{-8}$ | $3.6 \times 10^{-4}$ | $1.8 \times 10^{-4}$ | $1.9 \times 10^{-4}$ | $2.8 \times 10^{-1}$ |
| 2nd Quartile | $8.7 \times 10^{-7}$ | $1.1 \times 10^{-5}$ | $1.2 \times 10^{-3}$ | $1.2 \times 10^{-5}$ | $6.2 \times 10^{-8}$ | $1.6 \times 10^{-6}$ | $6.3 \times 10^{-5}$ | $3.9 \times 10^{-7}$ | $2.4 \times 10^{-6}$ | $6.7 \times 10^{-8}$ | $6.6 \times 10^{-5}$ | $2.4 \times 10^{-4}$ | $4.4 \times 10^{-4}$ | $3.4 \times 10^{-1}$ |
| 3rd Quartile | $1.1 \times 10^{-6}$ | $2.0 \times 10^{-5}$ | $2.8 \times 10^{-3}$ | $2.3 \times 10^{-5}$ | $9.2 \times 10^{-8}$ | $3.7 \times 10^{-6}$ | $9.3 \times 10^{-5}$ | $6.7 \times 10^{-7}$ | $7.9 \times 10^{-6}$ | $1.1 \times 10^{-7}$ | $7.5 \times 10^{-5}$ | $2.5 \times 10^{-4}$ | $5.7 \times 10^{-4}$ | $6.0 \times 10^{-1}$ |

## C   USAGE OF LARGE LANGUAGE MODELS (LLMS)

In this work, we used large language models (LLMs) for suggestions regarding refinement of writing, e.g. grammar, clarity and conciseness.

