# OpenReview forum: "PolySHAP: Extending KernelSHAP with Interaction-Informed Polynomial Regression"
_ICLR.cc/2026/Conference — ICLR 2026 Poster_

### Official Review · Reviewer_9go5 · 2025-10-28

**Soundness:** 2
**Presentation:** 2
**Contribution:** 2
**Rating:** 2
**Confidence:** 2

**Summary:**

The paper proposes an extension of KernelSHAP by fitting higher-order polynomials to approximate the Shapley value. The results show improvement with higher order polynomials. Interestingly, the authors find that fitting a 2nd order polynomial to approximate the Shapley value is equivalent to fitting KernelSHAP with paired (antithetic) sampling.

**Strengths:**

-The paper is proposing a novel idea to capture non-linear interactions between features.

-The authors provide theoretical support to the proposed idea.

**Weaknesses:**

-KernelSHAP with paired sampling has better convergence, and simply using the unbiased KernelSHAP [1], or vanilla KernelSHAP, with more data samples will improve the accuracy as well.

-In addition to the number of samples needed for PolySHAP, there is an additional cost to retrieve the Shapley values ($O(d·d')$), and this additional cost does not exist with KernelSHAP.

-KernelSHAP with paired sampling is equivalent to 2-PolySHAP, and k-PolySHAP representation is equivalent to order-k Faith-SHAP. Moreover, for tabular data, RegressionMSR generally outperforms PolySHAP. So why does the user need PolySHAP?

-In the experiments, the number of examples (10 randomly selected instances) is very small. I am not convinced that we can generalize the findings given this sample size.

-The unbiased KernelSHAP [1] is missing from the comparison.


[1]-Covert, I. and Lee, S.-I. Improving kernelshap: Practical Shapley value estimation using linear regression. In Proceedings of The 24th International Conference on Artificial Intelligence and Statistics, volume 130, pp. 3457–3465, April 2021.

**Questions:**

-Why widely used image datasets, e.g., CIFAR-10, CIFAR-100, and Imagenette, were not included in the experiments?

---

> ### Author Response · Authors · 2025-11-20
>
> We thank the reviewer for their time and their constructive feedback. We appreciate that they found our paper “novel” and “with theoretical support”, highlighting our paired sampling finding.
>
> > KernelSHAP with paired sampling has better convergence, and simply using the unbiased KernelSHAP [1], or vanilla KernelSHAP, with more data samples will improve the accuracy as well.
>
> As you point out, KernelSHAP with paired sampling is a better estimator than KernelSHAP without paired sampling. However, unbiased KernelSHAP is actually known to be worse than KernelSHAP, as noted in the paper that proposed it [1]. Nonetheless, we have added comparisons to unbiased KernelSHAP, see e.g., Figure 4 in the updated paper and the general statement.
>
> > In addition to the number of samples needed for PolySHAP, there is an additional cost to retrieve the Shapley values $O(d \cdot d’)$, and this additional cost does not exist with KernelSHAP.
>
> Since PolySHAP uses a more expressive function than KernelSHAP, it is true that solving the KernelSHAP regression problem is more computationally intensive. However, for the majority of machine learning applications, evaluating the game $\nu$ is the dominant cost.
>
> In Figure 14 in the updated paper, we compare the total cost of running each estimator (evaluation + regression) on various datasets. For CIFAR10 with a ViT16, for example, the **time is effectively the same for all estimators** because evaluating the value function takes so long. For other games that use more efficient tree-based games, the time complexity of RegressionMSR, KernelSHAP, 2-PolySHAP, 3-PolySHAP, and 4-PolySHAP are all close.
>
> > KernelSHAP with paired sampling is equivalent to 2-PolySHAP, and k-PolySHAP representation is equivalent to order-k Faith-SHAP. Moreover, for tabular data, RegressionMSR generally outperforms PolySHAP. So why does the user need PolySHAP?
>
> Great question! We believe the user benefits from PolySHAP for two reasons:
>
> 1. RegressionMSR is an accurate estimator *when* the gradient boosting tree accurately learns the game. However, this is not always the case e.g., **3PolySHAP outperforms RegressionMSR on the Housing, Adult, Estate, Forest, CG60, and Cancer datasets** (see e.g., Figure 4 and 7). As a result, we view PolySHAP as a tool in a user’s Shapley value estimation toolkit; when regressionMSR fails, they have another very accurate estimator.
>
> 2. A compelling advantage of PolySHAP is that it works with *any* set of interactions. For example, instead of using all order-3 interactions, we can use just a subset. In Figure 2, 4 and 7, for example, we highlight how using just a fraction of all order-3 interactions can lead to an improvement over KernelSHAP. We believe a particularly compelling avenue for future work is to carefully choose which higher order interactions to use in PolySHAP, balancing expressive power with the number of interaction terms. RegressionMSR already implicitly an interaction representation when building the XGBoost tree, and we suspect that carefully selecting the interactions for PolySHAP could further improve its performance.
>
> > In the experiments, the number of examples (10 randomly selected instances) is very small. I am not convinced that we can generalize the findings given this sample size.
>
> Evaluating the game is often the time bottleneck (e.g., when working with the ViT16). We have increased the number of repetitions to 30. **Notably, all of our findings with the increased repetitions are consistent with the first version of the paper.**
>
> > The unbiased KernelSHAP [1] is missing from the comparison.
>
> We have added unbiased KernelSHAP (which is actually equivalent to MSR) to our experiments in e.g., Figure 4, Figure 7, and the table below (see also general statement). As found in the original paper [1] and can be seen in our updated experiments, unbiased KernelSHAP returns much less accurate estimates than KernelSHAP.
>
> > Why widely used image datasets, e.g., CIFAR-10, CIFAR-100, and Imagenette, were not included in the experiments?
>
> We have added CIFAR-10 to our experiments using a fine-tuned ViT, we plan to add CIFAR-100 and Imagenette for the final version. We chose the non-tabular benchmarks according to the standardized and pre-computed benchmark games provided in the shapiq library (Muschalik et al., 2024).

---

> ### Author Response · Authors · 2025-11-20
>
> In the below table, we add the requested comparisons to SVARM, and MSR (which is equivalent to Unbiased KernelSHAP) for small and medium datasets, including the addition of the requested CIFAR-10 dataset. Our findings are consistent: Increasing the order of PolySHAP improves its performance. **Notably, 3-PolySHAP can give the lowest error on the Housing, Adult, Estate, Forest, CG60, and Cancer datasets (see e.g., Figure 4 and 7).**
>
> | approximator | Housing | ViT9 | Bike | Forest | Adult | DistilBERT | ResNet18 | Estate | CIFAR-10 ViT16 | ViT16 | Cancer |
>  |:----------------------|------------------:|---------------:|----------------:|------------------:|-----------------:|----------------------:|--------------------:|------------------:|--------------------------:|-----------------:|------------------:|
> | MSR                   |          0.000544 |       0.000115 |        2.06     |          0.0159   |         8.14e-07 |              3.37e-05 |            8.94e-06 |          0.0037   |                  0.000199 |         2.01e-05 |          2e-05    |
> | Unbiased KernelSHAP                   |          0.000544 |       0.000115 |        2.06     |          0.0159   |         8.14e-07 |              3.37e-05 |            8.94e-06 |          0.0037   |                  0.000199 |         2.01e-05 |          2e-05    |
> | SVARM                 |          0.000411 |       0.000101 |        2.21     |          0.0137   |         6.21e-07 |              3.08e-05 |            6.09e-06 |          0.00286  |                  7.29e-05 |         7.58e-06 |          4.69e-06 |
> | PermutationSampling   |          0.000164 |       0.000795 |        0.182    |          0.00419  |         9.23e-08 |              0.000127 |            2.04e-05 |          6.29e-05 |                  0.000323 |         2.77e-05 |          3.01e-07 |
> | KernelSHAP          |          7.83e-06 |       4.02e-05 |        0.0141   |          0.000945 |         6.08e-09 |              6.2e-06  |            1.78e-06 |          1.22e-05 |                  5.37e-05 |         4.58e-06 |          2.13e-07 |
> | RegressionMSR         |          1.04e-06 |       1.38e-05 |        0.000107 |          1.23e-06 |         8.58e-10 |              3.76e-06 |            1.02e-07 |          7.06e-08 |                  1.18e-05 |         1.38e-06 |          1.79e-08 |
> | PolySHAP-2ADD         |          7.83e-06 |       3.9e-05  |        0.0141   |          0.000945 |         5.97e-09 |              6.2e-06  |            1.78e-06 |          1.22e-05 |                  5.4e-05  |         4.56e-06 |          2.13e-07 |
> | PolySHAP-2ADD-50%     |          7.83e-06 |       3.89e-05 |        0.0139   |          0.00095  |         6.12e-09 |              6.2e-06  |            1.78e-06 |          1.22e-05 |                  5.39e-05 |         4.56e-06 |          2.13e-07 |
> | PolySHAP-3ADD         |          1.1e-07  |       3.15e-05 |        0.000147 |          1.18e-07 |         6.28e-11 |              2.53e-06 |            2.11e-07 |          9.86e-09 |                  1.51e-05 |         1.3e-06  |          1.3e-08  |
> | PolySHAP-3ADD-50%     |          9.24e-06 |       3.1e-05  |        0.0065   |          0.000957 |         8.71e-10 |              4.43e-06 |            1.09e-06 |          4.05e-06 |                  2.41e-05 |         2.21e-06 |          1.28e-07 |
> | PolySHAP-3ADD-dlog(d) |          1e-05    |       3.23e-05 |        0.0129   |          0.001    |         2.41e-09 |              5.85e-06 |            1.69e-06 |          4.38e-06 |                  3.7e-05  |         3.1e-06  |          1.7e-07  |
> | PolySHAP-4ADD         |          1.1e-07  |       3.11e-05 |        0.000147 |          1.17e-07 |         6.31e-11 |              2.54e-06 |            2.12e-07 |          1.03e-08 |                  1.51e-05 |         1.45e-06 |        nan        |

---

> ### Comment · Reviewer_9go5 · 2025-11-21
>
> Thank you very much for addressing my questions and providing clarifications. The additional empirical evidence has resolved most of my concerns. Therefore, I have accordingly raised my score for the paper.

---

### Official Review · Reviewer_ixVM · 2025-10-30

**Soundness:** 3
**Presentation:** 4
**Contribution:** 2
**Rating:** 4
**Confidence:** 3

**Summary:**

The paper introduces PolySHAP, a generalization of KernelSHAP that fits higher-order interaction terms through a polynomial (technically multilinear) regression model to better approximate the underlying Shapley game. This allows the method to capture non-additive feature interactions beyond what KernelSHAP can represent. The authors prove that PolySHAP yields consistent Shapley value estimates as the number of samples grows and establish a theoretical equivalence between KernelSHAP with paired sampling and second-order PolySHAP, providing an elegant explanation for the strong empirical performance of paired sampling. Experiments on various tabular, image, and text datasets demonstrate that higher-order PolySHAP variants improve accuracy over existing estimators, with paired KernelSHAP performing comparably to 2-PolySHAP but at a lower computational cost. Overall, the paper is clearly written, the experiments are extensive, and the connection between paired sampling and interaction modeling is both interesting and useful.

**Strengths:**

The paper is well written, conceptually clear, and provides a theoretical and empirical investigation of Shapley value estimation. The most notable strength is the discovery of a relationship between paired sampling and polynomial (interaction-based) fitting in KernelSHAP, which offers a new and elegant theoretical explanation for a long-standing empirical observation in the literature. This connection is both novel and insightful, bridging a practical heuristic with a principled mathematical foundation.

**Weaknesses:**

While I found the relation between paired sampling and polynomial (or mulilinear) fitting intriguing, I believe the PolySHAP formulation is not entirely novel.

The regression in Definition (4) corresponds exactly to the multilinear extension of cooperative games. Consequently, several theoretical results, including Theorem 4.2 follow directly from earlier works, particularly Owen’s papers on multilinear extensions [1] - See Section 2 for instance, and the Mobius representation of multilinear games that gives the same results as Theorem 4.2. This close relationship is not (explicitly) discussed in the paper.

That being said, the idea of using multilinear extensions in explainability is not entirely new. Methods such as Faith-SHAP already build on this formulation, and the equivalence stated in Corollary 4.5 seems straightforward in light of that prior work. There are also other recent studies exploring similar multilinear formulations for feature attribution [2] - they have an exact formulation of PolySHAP via the multilinear extension of games.


[1] https://www.jstor.org/stable/2661445
[2] https://ojs.aaai.org/index.php/AAAI/article/view/34149

**Questions:**

The model in Definition 4.1 effectively uses monomials (or multilinear extension) rather than full polynomials, which is the correct approach since for binary variables, higher-order powers are redundant. However, this point deserves explicit clarification, as using true polynomials would add unnecessary complexity without additional representational benefit.

---

> ### Author Response · Authors · 2025-11-20
>
> We thank the reviewer for their time and their constructive feedback. We appreciate that they found our paper “well written” and “conceptually clear”, highlighting our “new and elegant” explanation for the effectiveness of the practical paired sampling heuristic.
>
> > The regression in Definition (4) corresponds exactly to the multilinear extension of cooperative games. Consequently, several theoretical results, including Theorem 4.2 follow directly from earlier works, particularly Owen’s papers on multilinear extensions [1] - See Section 2 for instance, and the Mobius representation of multilinear games that gives the same results as Theorem 4.2. This close relationship is not (explicitly) discussed in the paper.
>
> Thank you for this comment! We would like to clarify that **we do not consider multilinear extensions of games.** The games we do consider are given by $\nu: 2^{[d]} \to \mathbb{R}$. Geometrically, we can think of this game as associating a value $\nu(S)$ with every corner $S \subseteq [d]$ of the boolean hypercube.
>
> The *multilinear extension* of a game extends it to the *interior* of the hypercube. Let $g : [0,1]^d \to \mathbb{R}$ be the (unique) multilinear extension given by
> $$
> g(\mathbf{x}) = \sum_{T \subseteq [d]} \nu(T) \prod_{i \in T} x_i \prod_{i \notin T} (1 - x_i)
> $$
> where $\mathbf{x} \in [0,1]^d$.
>
> We do *not* consider these multilinear extensions. Instead, the PolySHAP representation described in Definition 4.1 fits a *Möbius approximation* $\hat{v}:2^{[d]}$ to the game $\nu$ on a given set of interactions $\mathcal{I}$. The Möbius approximation is given by
> $$
> \hat{\nu}(S) = \sum_{T \in D \cup \mathcal{I}} \phi_T \prod_{j \in T} {1}[j \in S].
> $$
> This is a $\mathcal I$- restricted variant of the (exact) Möbius representation of the game
> $$
> \nu(S) = \sum_{T\subseteq D} \phi_T \prod_{j \in T}{1}[j \in S].
> $$
>
> You are absolutely right that there is a well-known result that we can obtain the Shapley values of $\nu$ using the formula in Eq. (3) and the true Möbius coefficients. Consequently, it is also obvious that the Shapley values of $\hat\nu$ are obtained by using Eq. (3) from our approximated Möbius coefficients $\phi_T$. However, Theorem 4.3 shows the surprising and novel result that the Shapley values of $\hat\nu$ are equal to the Shapley values of $\nu$ for **any restricted selection of interactions** $\mathcal I$, provided that we use the least-squares objective given by the PolySHAP representation to fit these (approximated) Möbius coefficients $\phi_T$ of $\hat\nu$. This is our crucial result that ensures that all PolySHAP estimates (independent of the choice of $\mathcal I$) converge to the true Shapley values. In fact, if other least-squares weights were used this would only hold if $\mathcal I$ would contain all exponentially many interaction terms.
>
> We respond to the remaining comments below.

---

> > ### Author Response · Authors · 2025-11-20
> >
> > > Methods such as Faith-SHAP already build on this formulation, and the equivalence stated in Corollary 4.5 seems straightforward in light of that prior work.
> >
> > The PolySHAP representation *is* closely related to Faith-SHAP, as highlighted in Corollary 4.5. However, Faith-SHAP is limited to interactions of a given size i.e., if we include an interaction of size $\ell$, Faith-SHAP must also include *all* interactions of size $\ell$. As a result, Faith-SAP is quite rigid and computationally intensive. In contrast, PolySHAP is flexible in that $\mathcal{I}$ can be *any* set of interactions. In particular, this opens the door to more computationally feasible approximations where we can consider only some terms of a higher order $\ell$.
> >
> > Moreover, the point of Faith-SHAP is to create a generalizations of Shapley values to higher order interactions (i.e., instead of returning the attribution of a single feature $i$, Faith-SHAP can return the attribution of sets of features $i,j,k$) but does not provide insights into the Shapley values of $\nu$. In contrast, we use the PolySHAP representation in a fundamentally different way:
> >
> > 1. Fit the Möbius approximation $\hat{\nu}$ to the game $\nu$ on sampled subsets.
> > 2. Return the Shapley values of $\hat{\nu}$ as our estimate for the Shapley values of $\nu$.
> > The justification behind our algorithm is the novel result described in Theorem 4.3: The Shapley values of the *best* PolySHAP approximation to $\nu$ are the Shapley values of $\nu$. In the language of statistical estimation, our approximation method is consistent i.e., if we sampled all subsets, then $\hat{\nu}$ would be the best approximation to $\nu$ and hence, by Theorem 4.3, its Shapley values would be the same as those of $\nu$.
> >
> > Theorem 4.3 was known for the special case of linear interaction terms i.e., KernelSHAP, and is trivially true if all exponential many interaction terms are included. But, to the best of our knowledge, it was *not* known for any selection of higher order interaction terms. On one hand, Theorem 4.3 is an independently interesting characterization of Shapley values. On the other hand, it also inspires the Shapley value estimation algorithm that we propose, which works quite well in practice, often beating the SOTA RegressionMSR for small and medium datasets.
> >
> >
> > > There are also other recent studies exploring similar multilinear formulations for feature attribution [2] - they have an exact formulation of PolySHAP via the multilinear extension of games.
> >
> > Thank you for bringing this work to our attention! As mentioned above, PolySHAP is actually fundamentally different because it is *not* a multilinear extension. However, we will be sure to cite this study in our revised related work section.

---

> > > ### Comment · Reviewer_ixVM · 2025-11-24
> > > **response - raise my score**
> > >
> > > I would like to thank the authors for their comprehensive response. After reviewing the response, the paper, the comments, and the rebuttal to other reviewers, I am convinced that the paper presents some novel ideas, and raise my score accordingly.

---

### Official Review · Reviewer_L6kD · 2025-10-30

**Soundness:** 2
**Presentation:** 2
**Contribution:** 2
**Rating:** 4
**Confidence:** 2

**Summary:**

This paper focuses on the Shapley value estimation problem in explainable artificial intelligence (XAI) and proposes a new method, **PolySHAP**, as an improvement over the widely used **KernelSHAP**. Traditional KernelSHAP employs a linear regression to approximate the relationship between model outputs and feature subsets, which limits its ability to capture feature interactions. This paper generalizes the approximation from a linear to a polynomial form, allowing the method to capture higher-order nonlinear interactions and improve the accuracy of feature attribution. In addition, the paper provides a theoretical explanation for the paired sampling mechanism in KernelSHAP, showing that it is equivalent to 2-PolySHAP.

**Strengths:**

* Presents a sound and theoretically grounded generalization of KernelSHAP by introducing polynomial regression to model nonlinear feature interactions, addressing a well-known limitation of additive interpretability methods.
* The connection between paired sampling and 2-PolySHAP is a notable theoretical insight, offering a formal explanation for an empirical heuristic widely used in practice.
* PolySHAP is shown to strictly subsume existing variants and guarantees convergence and consistency under reasonable assumptions.
* Supplementary experiments (Figures 4–5, Table 4) and ablation studies further support the robustness and generality of the proposed approach.

**Weaknesses:**

* **Limited scalability in high-dimensional or strongly non-additive settings.** Due to the combinatorial explosion, experiments with (k \geq 3) are restricted to small datasets, leaving open questions about PolySHAP’s feasibility in real-world large-scale scenarios.
* **Rapid growth in sampling and computational complexity.** The paper does not sufficiently discuss the trade-off between accuracy and computational efficiency when (k > 2).

**Questions:**

1. The paper states that KernelSHAP’s paired sampling is equivalent to 2-PolySHAP. Have the authors analyzed or compared the **computational efficiency** of the two methods? For example, do they exhibit similar runtime, sampling complexity, or scalability?
2. **Table 3** contains missing results. Could the authors clarify the reason for these omissions? From the current presentation, it appears that **3-PolySHAP becomes infeasible for large feature dimensions (d)**, suggesting scalability limitations. If so, what practical advantages does PolySHAP retain compared to KernelSHAP?
3. What was the **rationale behind the choice of sampling budgets and interaction orders** in the experiments? Were these hyperparameters tuned empirically, fixed a priori, or derived from theoretical considerations? Clarifying this would help assess the fairness and generality of the results.

---

> ### Author Response · Authors · 2025-11-20
>
> We thank the reviewer for their constructive feedback, and appreciation of our contribution regarding paired sampling and extensive empirical results.
>
> > Limited scalability in high-dimensional or strongly non-additive settings. Due to the combinatorial explosion, experiments with (k \geq 3) are restricted to small datasets, leaving open questions about PolySHAP’s feasibility in real-world large-scale scenarios. [...] Table 3 contains missing results. Could the authors clarify the reason for these omissions? From the current presentation, it appears that 3-PolySHAP becomes infeasible for large feature dimensions (d), suggesting scalability limitations. If so, what practical advantages does PolySHAP retain compared to KernelSHAP?
>
> The reason for the omissions in the table was that, for large datasets, there are too many terms if we use all or even half of order-3 interactions. To address your concern about “real-world large-scale scenarios”, we consider a 3-PolySHAP (log) variant in the rebuttal, that uses only a logarithmic number of the order-3 terms for higher dimensions. For small and medium datasets, **$3$-PolySHAP gives the best performance on the Housing, Adult, Estate, Forest, and Cancer datasets** (see Figure 4 and 7). For larger datasets, 3-PolySHAP (log) always outperforms KernelSHAP, and even RegressionMSR on the CG60 dataset.
>
> > Rapid growth in sampling and computational complexity. The paper does not sufficiently discuss the trade-off between accuracy and computational efficiency when (k > 2).
>
> The computational complexity of PolySHAP is discussed in Section 4.2. We have further extended the runtime analysis in Appendix B.5 and Figure 14. Previously, we only considered the tree-based benchmark games, where each evaluation only requires a single tree traversal. In this artificial setting, the additional computations in PolySHAP have an impact on the runtime of the approach. However, we added a ViT on CIFAR10 and show that the main driver of complexity for practical machine learning games are the model evaluations. In fact, evaluating the ViT on the sampled instances requires runtime of magnitudes higher than the computations used in PolySHAP or RegressionMSR. **In this practical example, the runtime of all estimators are effectively the same (see e.g., Figure 14).**
>
> > The paper states that KernelSHAP’s paired sampling is equivalent to 2-PolySHAP. Have the authors analyzed or compared the computational efficiency of the two methods? For example, do they exhibit similar runtime, sampling complexity, or scalability?
>
> Thank you for highlighting this important point! When using paired sampling, KernelSHAP (1-PolySHAP) and 2-PolySHAP result in identical estimates. However, 2-PolySHAP requires a larger budget than KernelSHAP for the regression problem to be determined, since the 2-PolySHAP uses more variables. In addition, solving the larger regression problem takes more time; although this time is often negligible in practice since the dominant runtime cost is evaluating the game (see e.g., Figure 14). We’d like to emphasize that we do *not* recommend using 2-PolySHAP; instead, we recommend using 3-PolySHAP (or its variants with fewer order-3 interactions). Notably, Figure 4 and Figure 7 indicate that 3-PolySHAP outperforms RegressionMSR on the Housing, Adult, Estate, Forest, CG60, and Cancer datasets.
>
> > What was the rationale behind the choice of sampling budgets and interaction orders in the experiments? Were these hyperparameters tuned empirically, fixed a priori, or derived from theoretical considerations? Clarifying this would help assess the fairness and generality of the results.
>
> We have chosen the sampling budget between $d+1$ to $20 000$ samples as related work uses similar ranges. We did not tune the number of interactions added in PolySHAP, and we did not make informed decisions about which interactions to choose. Instead, if the budget exceeded the number of regression variables, we included it in our benchmark, and we selected higher-order interactions in a random manner. In practice, we recommend to balance the number of interactions with the available budget $m$. However, we believe in future work the strongest benefits will be due to **structured selection** of interaction terms, e.g. as XGBoost implicitly does when fitting the surrogate in RegressionMSR. We leave the problem of selecting the most useful interaction terms to future work.

---

> > ### Comment · Reviewer_L6kD · 2025-11-25
> >
> > Thank you for your detailed reply. It addresses most of my earlier concerns. I now have one additional question. From your results and discussion, it appears that third-order interactions provide substantial benefits, whereas higher-order terms (such as fourth-order and above) do not seem to appear in your experiments. Should I interpret this as a practical limitation, meaning that modeling higher-order interactions becomes increasingly difficult or infeasible, and therefore their potential benefits cannot be reliably realized in PolySHAP?

---

> > > ### Author Response · Authors · 2025-11-25
> > >
> > > Thank you again for your question! We are glad that most of your earlier concerns are addressed. Your understanding is correct that, with paired sampling, order-3 interactions can yield a substantial benefit in practice. Under paired sampling, we actually observe that 3-PolySHAP is equivalent to 4-PolySHAP (see Figure 3 and “Higher Dimensional Extensions” on page 7), which is why we recommend using order-3 interactions and not order-4 interactions (similar to 1-PolySHAP vs. 2-PolySHAP). Using order-5 interactions can yield additional benefits but, due to the exploding number of interaction terms ($\binom{d}{5}=\mathcal O(d^5)$), we can rarely model all interactions, and the probability to randomly select the useful ones is small. However, our proposed PolySHAP representation can be used with any collection of interactions (some order-3, order-4, order-5, etc.), and our algorithm is still a consistent estimator of the Shapley values. If we choose interaction terms randomly (as in this work), then we recommend adding higher-order interactions only if lower-order interactions are fully covered. For future work, we believe that even stronger benefits can appear with **structured selection of interaction terms**, where higher-order interaction terms are carefully selected.

---

> > > > ### Comment · Reviewer_L6kD · 2025-11-27
> > > >
> > > > Thank you very much for the detailed clarification. This response has effectively resolved my remaining concerns, and the explanation regarding the practical behavior of higher-order interactions in PolySHAP is particularly helpful. I appreciate the additional insights into the equivalence between 3-PolySHAP and 4-PolySHAP under paired sampling, as well as the discussion on the feasibility of extending to higher-order terms.
> > > >
> > > > Overall, the revisions and explanations have substantially improved my understanding of the method. I will update my evaluation accordingly.

---

### Official Review · Reviewer_cf9r · 2025-10-31

**Soundness:** 3
**Presentation:** 3
**Contribution:** 3
**Rating:** 8
**Confidence:** 3

**Summary:**

The paper outlines theoretical and practical analysis of a method of obtaining more accurate shapley value estimates. The method essentially recognizes that shapley values are linear least square coefficients, so it computes higher-order pynomial leaest-sqaure estimates, then distributes higher-order coefficients down to individual variables using the shapley value-value property for projecting pure interaction functions. Comparison/euqivalencies to existing shapley work, including paired kernelShap, are given, as well as emperical convergence analysis.

**Strengths:**

- relevant theoretical results
- sound theoretical basis
- useful experimental results
- reasonably clear

**Weaknesses:**

- I expect more experinental comparisons to other existing shap-acceleration methods in terms of speed and accuacy, as such, it is difficult to assess the effectiveness of the method.

**Questions:**

- Please comment on the results of the experimetns sectin, as it appears that regression MSR outperforms polySHAP.
- Have you considered applying the approximate-up-then-project-down approach to other shapley-like methods that are computationally expensive and require random sampling, such as the Shapley-Taylor Interaction Index?

small issues:
- table 1 needs more visual separation from paragraph text
- 228 - "this is approach is"

---

> ### Author Response · Authors · 2025-11-20
>
> We thank the reviewer for their appreciation of our theoretical results, and constructive feedback on improving the empirical part of our work.
>
> > I expect more experinental comparisons to other existing shap-acceleration methods in terms of speed and accuacy, as such, it is difficult to assess the effectiveness of the method.
>
> In terms of accuracy, we have added MSR, SVARM, and Unbiased KernelSHAP baselines to our experiments and provided the new comparison in a separate Figure 4 and Figure 7 (described in the general statement). Our empirical results confirm previous studies by Musco & Witter (2025) and Witter, Liu, & Musco (2025) that KernelSHAP with leverage score sampling and RegressionMSR provide state-of-the-art performance, followed by Permutation Sampling. Notably, Figure 4 and Figure 7 indicate that 3-PolySHAP outperforms RegressionMSR on the Housing, Adult, Estate, Forest, CG60, and Cancer datasets.
>
> In terms of time, the dominant cost for the majority of machine learning applications is evaluating the game $\nu$. In Figure 14 in the updated paper, we compare the total cost of running each estimator (evaluation + regression) on various datasets. For CIFAR10 with a ViT16, for example, the **time is effectively the same for all estimators** because evaluating the value function takes so long. For artificial tree-based games, the time complexity of RegressionMSR, KernelSHAP, 2-PolySHAP, 3-PolySHAP, and 4-PolySHAP are all close (see e.g., Figure 14).
>
> > Please comment on the results of the experimetns sectin, as it appears that regression MSR outperforms polySHAP.
>
> Great question! We have two comments:
> 1. RegressionMSR is an accurate estimator *when* the gradient boosting tree accurately learns the game. However, this is not always the case e.g., **3-PolySHAP outperforms RegressionMSR on the Housing, Adult, Estate, Forest, CG60, and Cancer datasets** (see e.g., Figure 4 and 7). As a result, we view PolySHAP as a tool in a user’s Shapley value estimation toolkit; when RegressionMSR is less accurate, they have another very accurate option.
> 2. A compelling advantage of PolySHAP is that it works with *any* set of interactions. For example, instead of using all order-3 interactions, we can use just a subset. In Figure 2, 4 and 7, for example, we highlight how using just a fraction of all order-3 interactions can lead to an improvement over KernelSHAP. We believe a particularly compelling avenue for future work is to carefully choose which higher order interactions to use in PolySHAP, balancing expressive power with the number of interaction terms. RegressionMSR already implicitly fits an interaction representation when building the XGBoost tree, and we suspect that carefully selecting the interactions for PolySHAP could further improve its performance.
>
> > Have you considered applying the approximate-up-then-project-down approach to other shapley-like methods that are computationally expensive and require random sampling, such as the Shapley-Taylor Interaction Index?
>
> Great question! We have some initial thoughts: The Shapley Taylor interaction index and the n-Shapley values (Bordt & von Luxburg, 2023) can be obtained similarly to the PolySHAP representation from other interaction indices. However, there isn’t a least-squares representation for these indices, so it’s unclear how to define the projection. The leading approximation methods for the Shapley Taylor indices and n-Shapley values (extensions of MSR, SVARM, Permutation Sampling) compute each interaction in an isolated manner using Monte Carlo sampling. We thus suspect that the two-step approach from PolySHAP will not yield similar benefits, since information is not shared between estimated interaction terms as in our least-squares objective. However, we leave further investigation to future work.

---

### Author Response · Authors · 2025-11-20
**General Rebuttal Comment**

We thank the reviewers for their time and constructive feedback on our work. Overall, we highly appreciate the consensus on the value of our theoretical insights into paired sampling and the consistent Shapley value estimator we propose. According to the feedback we received, we identified potential to improve the empirical part of our work as follows:

**Adding more baselines**: We added SVARM (Kolpaczki et al., 2024), MSR (Wang & Jia, 2023) and Unbiased KernelSHAP (Covert & Lee, 2021) to our experiments. Confirming previous studies (Covert & Lee, 2021 and Musco & Witter, 2025), we find that these baselines are already substantially outperformed by KernelSHAP (1-PolySHAP) with leverage score sampling. The baseline comparison can now be found in a new Figure 4 and Figure 7.

**Practical Benefits of PolySHAP**: We added Figure 4 and Figure 7 to compare PolySHAP in a realistic setting under paired sampling with all baselines, and added a paragraph to the main text (“Practical Benefits of PolySHAP”) discussing the potential of PolySHAP under paired sampling:

> In practice, we adopt paired sampling and benchmark PolySHAP against all baselines in Figure 4 and Figure 7. Because of our paired sampling result, the practical benefits of PolySHAP become apparent only when order-$3$ interactions are included. In low-dimensional settings, the $3$-PolySHAP yields the best performance on Housing, Adult, Estate, Forest, and Cancer datasets (see e.g., Figure 4 and 7). In budget-restricted cases, partially incorporating order-$3$ interactions already provides substantial gains, cf. $3$-PolySHAP (50\%) and $3$-PolySHAP (log). In high-dimensional settings ($d\geq 60$), however, only a small number of order-$3$ interactions can be added, resulting in more modest improvements. Among all baselines, only RegressionMSR achieves comparable performance, although its performance depends strongly on XGBoost, as indicated by its poor results on CG60. Moreover, RegressionMSR has an inherent advantage since all tabular games rely on tree-based models.

Addressing concerns about the high dimensional setting, we added a variant that uses only a logarithmic number of order-3 terms. The 3-PolySHAP (log) algorithm allows for order-$3$ interactions in high-dimensional settings, outperforming paired KernelSHAP. (We also moved Table 4 to the appendix, since it gave only insights for a single choice of budget.)

**Extension of Experiments**: As requested by Reviewer 9go5, we added a CIFAR10 game using fine-tuned ViT pre-trained on ImageNet and fine-tuned on CIFAR10. Moreover, we increased the number of explained instances from $10$ to $30$, which yields even smaller error bars, but does not change the initial empirical results.

**Runtime Analysis**: We extended the runtime analysis in Appendix B.5 with the real-world application setting of the CIFAR10 game. As expected, we observe that the main driver for computational complexity are the model calls (game evaluations), which is magnitudes higher than the computations in PolySHAP. In contrast, the runtime analysis on the tabular tree games, where evaluations only require a single tree traversal, confirms the polynomial scaling in terms of modeled interaction terms.

---

### Author Response · Authors · 2025-11-29
**Pre-Leak Reviewer Engagement Timeline**

We thank the reviewers for their constructive feedback and engagement. We have learned that scores will be rolled back due to an OpenReview bug reported at *7:09am PT on November 27*. Because we believe the reviewers’ comments and score changes *prior to the bug* are important, we highlight their timeline below:

Reviewer 9go5 responded at *1:14am PT on November 21*, **increasing their score from 2 to 6.**

> Thank you very much for addressing my questions and providing clarifications. The additional empirical evidence has resolved most of my concerns. Therefore, I have accordingly raised my score for the paper.

*(Posted more than 6 days before the bug was reported.)*

Reviewer ixVM responded at *2:01am PT on November 24*, **increasing their score from 4 to 8.**

> I would like to thank the authors for their comprehensive response. After reviewing the response, the paper, the comments, and the rebuttal to other reviewers, I am convinced that the paper presents some novel ideas, and raise my score accordingly.

*(Posted more than 3 days before the bug was reported.)*

Reviewer L6kD asked a question at *11:52pm PT on November 24*, we replied at *7:24am PT on November 25*, and Reviewer L6kD responded at *10:11pm PT on November 26*, **increasing their score from 4 to 6.**

> Thank you very much for the detailed clarification. This response has effectively resolved my remaining concerns, and the explanation regarding the practical behavior of higher-order interactions in PolySHAP is particularly helpful. I appreciate the additional insights into the equivalence between 3-PolySHAP and 4-PolySHAP under paired sampling, as well as the discussion on the feasibility of extending to higher-order terms. Overall, the revisions and explanations have substantially improved my understanding of the method. I will update my evaluation accordingly.

*(Posted about 8 hours before the bug was reported.)*

Overall, **our *pre-bug* responses addressed reviewers’ major concerns, as reflected in the average reviewer score increasing from 4.5 to 7.**

---

### Meta-Review · Area_Chair_ssGK · 2026-01-04

**Summary:**

This paper proposes PolySHAP, a generalization of KernelSHAP that fits higher-order (multilinear) regression models to more accurately estimate Shapley values and capture feature interactions. Reviewers agree that the paper is theoretically grounded, clearly written, and addresses an important problem in explainable AI. Reviewers acknowledged a key insight from the paper that KernelSHAP with paired sampling is equivalent to 2-PolySHAP, which provides a formal explanation for a widely used empirical observation. The paper provides convergence analysis and experimental results supporting improved estimation accuracy under appropriate settings. Overall, the work makes a meaningful contribution by unifying existing Shapley estimation techniques using a principled framework.

**Reviewer Concerns:**

The main concerns in the original reviews focused on the experimental scope, the feasibility of extending to higher-order terms, and the clarification on higher-order interactions and connections with prior work (multilinear extensions in explainability, and related shap-acceleration methods). In the rebuttal, the authors clarified the positioning of PolySHAP relative to existing methods, explained design choices and experimental settings, and addressed questions about experimental scope with additional results. Reviewers replied that their concerns were adequately addressed and acknowledged that contribution of this work.

**Reviewer Scores:**

Reviewer cf9r gave an original score of 8, and their original questions are appropriately addressed. I'd expect that their final score to remain positive.

The other three reviewers gave original negative scores, but they've explicitly stated in the response that the rebuttal addressed their concerns. I'd expect that their final scores to be positive.

---

### Decision · Program_Chairs · 2026-01-26

Accept (Poster)